# EQA-MX: Embodied Question Answering using Multimodal Expression

**Md Mofijul Islam**[*][‡][§] **, Alexi Gladstone**[‡]**, Riashat Islam**[†]**, Tariq Iqbal**[‡]
[‡] University of Virginia, [§] Amazon GenAI, [†] McGill University, Mila, Quebec AI Institute

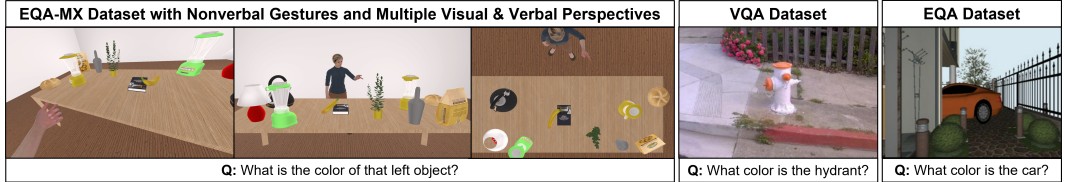

Figure 1: Compared to the QA tasks in existing VQA (Antol et al., 2015) and EQA (Das et al., 2018a) datasets, the models to answer EQA tasks in our EQA-MX dataset require the reasoning of questions with multimodal expressions (verbal and nonverbal gestures).

## Abstract

Humans predominantly use verbal utterances and nonverbal gestures (e.g., eye gaze and pointing gestures) during natural interactions. For instance, pointing gestures and verbal information is often required to comprehend questions such as "what object is that?" Thus, this question-answering (QA) task involves complex reasoning of multimodal expressions (verbal utterances and nonverbal gestures). However, prior works have explored QA tasks in non-embodied settings, where questions solely contain verbal utterances from a single verbal and visual perspective. In this paper, we have introduced 8 novel embodied question answering (EQA) tasks to develop learning models to comprehend embodied questions with multimodal expressions. We have developed a novel large-scale dataset, EQA-MX, with over 8 million diverse embodied QA data samples involving multimodal expressions from multiple visual and verbal perspectives. To learn salient multimodal representations from discrete verbal embeddings and continuous wrappings of multiview visual representations, we propose a vector-quantization (VQ) based multimodal representation learning model, VQ-Fusion, for EQA tasks. Our extensive experimental results suggest that VQ-Fusion can improve the performance of existing visual-language models up to 13% across EQA tasks.

## 1 Introduction

Understanding human instructions is crucial for autonomous agents to effectively collaborate with humans (Chen et al., 2021; Kratzer et al., 2020; Islam et al., 2022b;a). To develop models for instruction comprehension, several tasks have been designed, such as referring expression comprehension (Yang et al., 2019b; Yu et al., 2016; Kamath et al., 2021; Akula et al., 2021; Chen et al., 2020a), spatial relations grounding (Yang et al., 2019a; Viethen & Dale, 2008; Achlioptas et al., 2020; Liu et al., 2022), and visual question answering (Antol et al., 2015; Gao et al., 2015; Yu et al., 2015; Zhu et al., 2016; Krishna et al., 2017; Kafle et al., 2018; Gurari et al., 2018). Notably, VQA has gained significant attention due to its complex reasoning demands, such as answering questions about object presence and category using visual cues (Antol et al., 2015; Goyal et al., 2017; Lee et al., 2022).

Numerous synthetic and real-world datasets exist for VQA, yet their sole focus on verbal questions is a crucial limitation, contrasting with natural multimodal expressions (verbal utterances and nonverbal gestures) used in inquiries. Studies affirm that nonverbal gestures often provide complementary information for understanding verbal questions (McNeill, 2012; Corkum & Moore, 1998; Butterworth et al., 2002; Scaife & Bruner, 1975; Colonnesi et al., 2010; Iverson & Goldin-Meadow, 2005; Kita, 2003; Liszkowski et al., 2004; Chen et al., 2021). For example, in a scene with two differently

---

[*]Work does not relate to position at Amazon.

Table 1: Comparison of the QA datasets. Existing VQA and EQA datasets do not contain nonverbal gestures (NV), multiple verbal (V) perspectives (MVP), contrastive (C) and ambiguous (A) data samples. [‡] Embodied (E) interactions refer to humans interacting using multimodal expressions. [†] Embodied interactions refer to an agent navigating in an environment. **Please check the supplementary for a detailed comparison with other related datasets.**

| Datasets | V | NV | E | EQA | MVP | Views | | | C | A | No. of Images | No. of Samples |
|---|---|---|---|---|---|---|---|---|---|---|---|---|
| | | | | | | Exo | Ego | Top | | | | |
| VQA (Antol et al., 2015) | ✓ | ✗ | ✗ | ✗ | ✗ | ✓ | ✗ | ✗ | ✗ | ✗ | 204k | 614k |
| KB-VQA (Wang et al., 2015) | ✓ | ✗ | ✗ | ✗ | ✗ | ✓ | ✗ | ✗ | ✗ | ✗ | 0.7k | 5k |
| FBQA (Wang et al., 2017) | ✓ | ✗ | ✗ | ✗ | ✗ | ✓ | ✗ | ✗ | ✗ | ✗ | 2k | 5k |
| VQA-MED (Hasan et al., 2018) | ✓ | ✗ | ✗ | ✗ | ✗ | ✓ | ✗ | ✗ | ✗ | ✗ | 2k | 6k |
| DocVQA (Mathew et al., 2021) | ✓ | ✗ | ✗ | ✗ | ✗ | ✓ | ✗ | ✗ | ✗ | ✗ | 12k | 50k |
| GRiD-3D (Lee et al., 2022) | ✓ | ✗ | ✗ | ✗ | ✗ | ✓ | ✗ | ✗ | ✗ | ✗ | 8k | 445k |
| VIMA (Jiang et al., 2022) | ✓ | ✗ | ✗ | ✗ | ✗ | ✓ | ✗ | ✗ | ✗ | ✗ | 650k | 650k |
| EQA [†] (Das et al., 2018a) | ✓ | ✗ | ✓[†] | ✓[†] | ✗ | ✗ | ✓[†] | ✗ | ✗ | ✗ | 5k | 5k |
| MT-EQA [†] (Das et al., 2018a) | ✓ | ✗ | ✓[†] | ✓[†] | ✗ | ✗ | ✓[†] | ✗ | ✗ | ✗ | 19k | 19k |
| EQA-MX [‡] | ✓ | ✓ | ✓ | ✓ | ✓ | ✓ | ✓ | ✓ | ✓ | ✓ | 750k | 8,243k |

colored balls, a pointing gesture can clarify questions like "what is the color of that ball?" The absence of nonverbal interactions in prior VQA datasets makes them less suitable for developing models to comprehend question-answering (QA) tasks in embodied settings.

Following VQA, embodied question-answering (EQA) tasks have recently been studied in the literature (Yu et al., 2019; Luo et al., 2019; Gordon et al., 2018; Tan et al., 2020). EQA can be bifurcated based on embodied interactions: the first centers on an agent, like a virtual robot, navigating to answer verbal questions (Das et al., 2018a), solely incorporating verbal queries. The second encompasses multimodal expressions, where humans interact with the environment using verbal utterances and gestures (Chen et al., 2021; Islam et al., 2022a). Adopting the latter definition, we designed EQA tasks to comprehend questions using multimodal expressions (verbal uttrances and nonverbal gestures) in embodied settings. For instance, an EQA task may involve pointing to an object and asking "what is that object?" requiring reasoning over multimodal expressions to answer the question.

A notable limitation in many existing VQA and EQA datasets is the singular perspective (either speaker or observer) of verbal utterances, unlike real-world interactions where where people use both perspective interchangeably. For instance, a speaker's question, "What is the object to the *right of the red mug*?" could be interpreted as *left of the red mug* from an observer's perspective. This lack of multiple perspectives in existing datasets hinders the development of robust QA models.

Similarly, existing VQA and EQA models answer questions from a single visual perspective (Li et al., 2019; Kim et al., 2021; Lu et al., 2019). Multiple views provide complementary information, and varying camera angles capture interactions differently. Aligning visual representations before merging with verbal ones can aid in developing generalized representations and robust comprehension across perspectives. Moreover, the inconsistency of embedding structures, particularly continuous visual and discrete verbal representations, can lead to sub-optimal representations.

To address the shortcomings of existing VQA and EQA datasets, we have extended an embodied simulator to develop a large-scale novel dataset, EQA-MX, for comprehending EQA tasks (Table 9). We have addressed the limitations of existing multimodal fusion approaches and developed a multimodal learning model for EQA tasks, VQ-Fusion, using vector quantization (VQ). The VQ-based bottleneck plays a key role in disentangling the continuous visual representations into discrete embeddings and enables salient fusion with discrete verbal representations. We use a shared codebook in VQ to align multiview representations and learn the unified concept shared among multiple views. We highlight our *key contributions* below:

- We developed a large-scale dataset (EQA-MX) with multimodal expressions from various verbal and visual perspectives to reduce perspective bias and enhance model generalizability.
- We designed 8 new EQA tasks blending multimodal questions (verbal and gestural) to be addressed using visual context in an embodied setting.
- We designed a VQ-based multimodal fusion method to align continuous visual and discrete verbal representations and extract salient representations across multiple visual and verbal perspectives.
- Our extensive experimental analyses indicate that our proposed model, VQ-Fusion, can help to improve the performance of EQA tasks up to 13%.

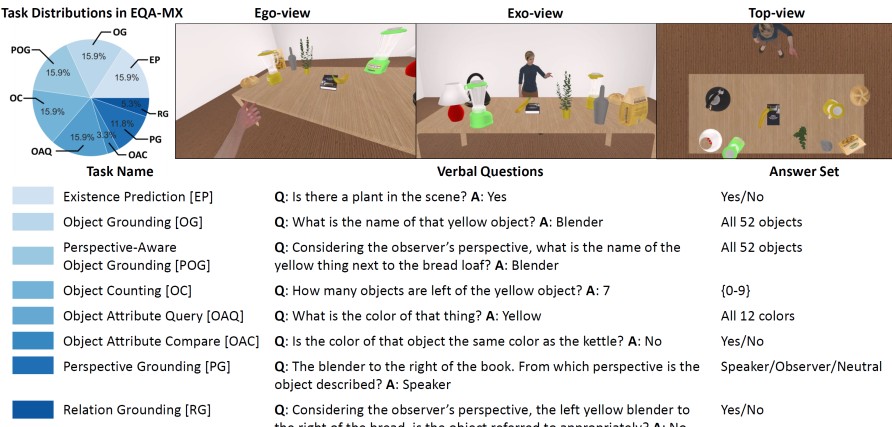

| Task Name | Verbal Questions | Answer Set |
|---|---|---|
| Existence Prediction [EP] | **Q**: Is there a plant in the scene? **A**: Yes | Yes/No |
| Object Grounding [OG] | **Q**: What is the name of that yellow object? **A**: Blender | All 52 objects |
| Perspective-Aware Object Grounding [POG] | **Q**: Considering the observer's perspective, what is the name of the yellow thing next to the bread loaf? **A**: Blender | All 52 objects |
| Object Counting [OC] | **Q**: How many objects are left of the yellow object? **A**: 7 | {0-9} |
| Object Attribute Query [OAQ] | **Q**: What is the color of that thing? **A**: Yellow | All 12 colors |
| Object Attribute Compare [OAC] | **Q**: Is the color of that object the same color as the kettle? **A**: No | Yes/No |
| Perspective Grounding [PG] | **Q**: The blender to the right of the book. From which perspective is the object described? **A**: Speaker | Speaker/Observer/Neutral |
| Relation Grounding [RG] | **Q**: Considering the observer's perspective, the left yellow blender to the right of the bread, is the object referred to appropriately? **A**: No | Yes/No |

Figure 2: EQA tasks for a sample data from EQA-MX. Top-row: data distribution for each task in EQA-MX (left) and an embodied interaction with multiple visual perspectives (right). Bottom-row: name of the task (left), example questions and answers for the given task based on the visual scene above (middle), and the set of possible answers (right).

## 2 RELATED WORK

**Visual Question Answering:** Many datasets have been developed to study visual question-answering tasks (Gao et al., 2015; Zhu et al., 2016; Liu et al., 2019; Krishna et al., 2017; Wang et al., 2015; 2017; Kembhavi et al., 2016; Kahou et al., 2017; Kafle et al., 2018; Gurari et al., 2018; Hasan et al., 2018; Huang et al., 2018; Andreas et al., 2016; Chou et al., 2020; Hudson & Manning, 2019; Mishra et al., 2019). These datasets primarily involve answering verbal questions using the visual scene as context. For example, Antol et al. (2015) developed a VQA dataset and introduced QA tasks involving an image and verbal questions about the image. This dataset contains both real-world images from the MS-COCO dataset (Lin et al., 2014) as well as synthetic virtual scenes containing clipart. Ren et al. (2015) generated synthetic QA pairs using an algorithm that converts image descriptions into QA form. Recently, a few datasets have been developed containing multimodal expressions (Schauerte & Fink, 2010; Islam et al., 2022b). For example, Chen et al. (2021) developed a dataset for referring expression comprehension tasks in embodied settings, where a human uses multimodal expressions to refer to an object.

Several visual-language (VL) models have been developed for VQA tasks and were consequentially evaluated on these datasets (Radford et al., 2021; Lu et al., 2019; Tan & Bansal, 2019; Chen et al., 2020a). For example, Liunian et al. (Li et al., 2019) developed VisualBERT to answer a question using the visual context by learning multimodal representations from visual and verbal embeddings. Kim et al. (Kim et al., 2021) designed a VL Transformer model (ViLT) with monolithic processing of visual inputs to learn VL representations without regional supervision of object detection.

**Embodied Question Answering (EQA):** EQA tasks are often designed as agents (e.g., virtual robots) navigating in an environment to answer a verbal question. For example, Das et al. (2018a) developed a synthetic dataset, where a virtual robot navigates the environment and gathers visual information from an egocentric view to answer a verbal question. Yu et al. (2019) extend this dataset and include questions with multiple targets, such as finding multiple objects through navigation. However, some works have used embodied interactions to refer to comprehending referring expressions (Islam et al., 2022b; Chen et al., 2021). We follow this definition of embodied interaction.

Several models have been developed for existing EQA tasks. For instance, Das et al. (2018b) introduced a modular model for learning a policy to navigate and answer verbal questions, while Gao et al. (2021) utilized a transformer-based model to generate scene memory tokens as exploration clues. These models aim to develop a navigation policy for answering verbal questions.

Most current VQA and EQA studies focus on understanding solely verbal questions, contrasting our goal of comprehensively understanding multimodal expressions (verbal utterances and gestures) in embodied settings. Moreover, existing models fuse disparate embedding structures (continuous visual and discrete verbal representations), potentially leading to sub-optimal VL representations.

# 3   EMBODIED QUESTION ANSWERING TASKS

We have created 8 novel EQA tasks: Existence Prediction (EP), Object Grounding (OG), Perspective-Aware Object Grounding (POG), Object Counting (OC), Object Attribute Query (OAQ), Object Attribute Compare (OAC), Perspective Grounding (PG), and Relation Grounding (RG). Similar tasks have been developed in prior works (Antol et al., 2015; Lee et al., 2022; Wang et al., 2017; Yu et al., 2016; Ren et al., 2015; Zhu et al., 2016), however, those tasks involve only verbal questions. We are the first to design QA tasks in embodied settings where a human avatar asks questions using verbal utterances and nonverbal gestures in a virtual environment. Each task has multiple sub-templates for variation (described further in the supplementary materials). In Fig. 2, we provide samples of these EQA tasks.

**Existence Prediction (EP):** The EP task involves determining whether the scene contains a particular object with specific attributes (e,g., color, location). Completing this task requires knowledge of object appearances as well as a holistic understanding of the scene.

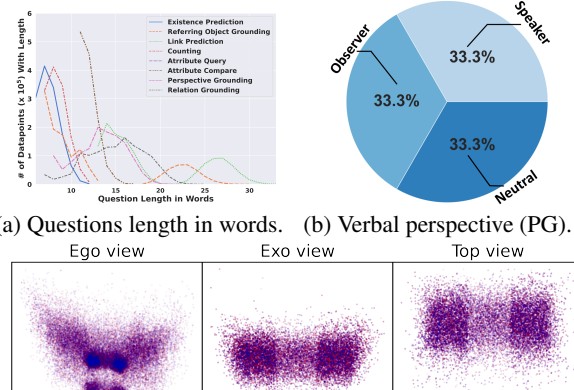

(a) Questions length in words.   (b) Verbal perspective (PG).

(c) Object locations with respect to spatial relations.

Figure 3: EQA-MX Dataset Analysis: (a) demonstrates varied question lengths in the EQA-MX dataset, indicating differing contextual information across EQA tasks. (b) presents data sample ratios of different verbal perspectives for the perspective grounding (PG) task. (c) depicts object locations in relation to different spatial relations, showing the EQA-MX dataset's non-bias towards verbal and visual perspectives due to inseparable object locations. **For detailed analysis, refer to the supplementary materials.**

**Object Grounding (OG):** In the OG task, the object category is determined based on the question by utilizing multimodal expressions. This task also involves understanding which perspective the question is asked from (i.e., speaker, observer, neutral).

**Perspective-Aware Object Grounding (POG):** Similar to the OG task, the POG task involves determining which object is being referred to. However, this task also includes the verbal perspective in the question (speaker, observer, neutral). This is done intentionally to determine how much of an impact verbal perspective has on performance.

**Object Counting (OC):** In the OC task, the number of objects in a scene is asked based on different spatial relations. To understand this, different objects in the visual scene must be attended to and spatial relations given in the verbal question must be used to determine whether or not certain objects have that attribute prior to counting.

**Object Attribute Query (OAQ):** The OAQ task involves determining the color of a given object that is queried for, which can be helpful in scenarios where humans are interested in learning particular characteristics of an object. The spatial location and the color of the object must be determined using the given verbal and nonverbal expressions.

**Object Attribute Compare (OAC):** The OAC task entails comparing two objects' attributes, involving pointing to an object and querying their similarity in attributes.

**Perspective Grounding (PG):** Understanding human verbal perspective is crucial for effective human-AI communication, as humans describe objects from varying perspectives. We simulate this by employing three perspectives - neutral, speaker, and observer, tasking the model with identifying the perspective of a given question.

**Relation Grounding (RG):** The RG task involves determining whether a verbal utterance and nonverbal gestures refer to the same object. As a referring expression can be interpreted differently from different visual and verbal perspectives, understanding the RG task requires complex reasoning of perspective and spatial relations.

## 4 DATASET GENERATION WITH EQA SIMULATOR

In this work, we have extended the CAESAR simulator (Islam et al., 2022b) to generate data for different EQA tasks. CAESAR is used to randomly generate environments where an actor simulates nonverbal expressions through a pointing gesture and gaze in a scene (Fig. 3). Verbal expressions are created based on the visual scene. To increase the dataset's generalizability, we have used multiple environments. These environments differ in terms of camera views, object locations, and nonverbal/verbal expressions. In each visual scene, we generated four different situations, 1) a situation with no human and therefore no nonverbal expressions, 2) a situation with a human head gaze, 3) a situation with a human pointing gesture, and 4) a situation involving a human using a head gaze and a pointing gesture.

Generated nonverbal expressions consist of a pointing gesture and gaze. Pointing gestures are procedurally generated using inverse kinematics through the Unity engine. We create these pointing gestures based on random noise added onto real-world data of human pointing gestures captured using an Optitrack motion capture system (opt). Similarly, we have simulated human head gazes using inverse kinematics and an object location within the scene as a target. Verbal questions are generated based on different templates for each EQA task. The nonverbal and verbal expressions may describe the same object, or be contrastive, meaning the nonverbal and verbal expressions describe different objects. We use these contrastive instructions for the Relation Grounding task. Additionally, the absence of nonverbal gestures in situations with no humans generates ambiguous data samples. Please check the supplementary document for additional details on the data generation process.

## 5 DATASET ANALYSIS

We have generated a novel large-scale dataset, EQA-MX, containing $8,243,893$ samples across the 8 tasks described in Sect. 3. The training,

Table 2: EQA-MX dataset splits for 8 EQA tasks.

| Splits | EP | OG | POG | OC | OAQ | OAC | PG | RG |
|---|---|---|---|---|---|---|---|---|
| Train | 1060k | 1060k | 1060k | 1060k | 1060k | 218k | 785k | 349k |
| Valid | 126k | 126k | 126k | 126k | 126k | 27k | 93k | 41k |
| Test | 126k | 126k | 126k | 126k | 126k | 28k | 93k | 42k |

validation, and test set splits for each of these tasks is shown in Table 2. We removed some data samples to generate balanced dataset splits for the OAC, PG, and RG tasks.

Our designed EQA tasks vary in terms of the goals (Fig. 2) and visual-verbal contextual information in the questions. This is made apparent by the variance in question lengths in words (Fig. 3(a)). Questions are as short as 6 words for the EP task and as long as 34 words for the OG task. Additionally, one of the main focuses of the EQA-MX dataset is to introduce data that varies in verbal and visual perspectives. Fig. 3(b) demonstrates the PG task's outcome of different verbal perspectives.

Similarly, Fig. 3(c) shows the location of objects based on spatial relations in questions from verbal perspectives. Fig. 3(c) also demonstrates how objects being referred to as on the left (blue) and right (red) are not linearly separable through the use of spatial relations, as different verbal perspectives use different relations to describe an object. For example, consider a speaker describing the red table lamp in Fig. 2. The speaker could state "the red lamp on the left". However, from the observer's perspective (exo view) the table lamp is on the right. Thus, given the verbal perspectives, spatial relations are non-separable in EQA-MX (Fig. 3(b)). This reduced verbal and visual perspective biases in EQA-MX dataset can help train robust models for comprehensively comprehending EQA tasks. **Please check the supplementary materials for a more detailed data analysis.**

## 6 VQ-FUSION: VQ-BASED MULTIMODAL FUSION

We develop a vector quantization-based multimodal fusion approach, VQ-Fusion, to learn visual-language representations. As EQA tasks in EQA-MX involve multiple visual views, VQ-Fusion extracts visual representations from multiple visual views ($X_{ego}$, $X_{exo}$, and $X_{top}$) and verbal questions ($X_q$) for different EQA tasks (Fig. 4 and Sect. 3). Following the existing adapter-based learning models (Beck et al., 2022; Ansell et al., 2021; Rücklé et al., 2021; Pfeiffer et al., 2020a;b; 2021), we design VQ-Fusion as an adapter model that can be used in existing models without significantly changing the existing model architecture.

**Visual and Language Representation Learning:** At first, VQ-Fusion extracts visual and language representations using a state-of-the-art visual encoder (e.g., ResNet (He et al., 2016) and

ViT (Dosovitskiy et al., 2020)) and language model (e.g., BERT (Devlin et al., 2018)). VQ-Fusion uses shared models to extract the visual representations from multiple views independently: $E_m = F_m(X_m), m \in (ego, exo, top, verbal)$. Here, $F_m$ is the visual or verbal encoders, $E_m \in \mathbb{R}^{D_m}$, and $D_m$ is the representation dimension of modality $m$.

**Discretization and Multimodal Fusion:**
Language models create discretized representations, whereas visual encoders produce continuous representations of visual scenes. Fusing these representations with different embedding structures can lead to sub-optimal multimodal representations (Liang et al., 2022). For this reason, we discretize the visual representations before fusion.

In VQ-Fusion, we adopted the vector quantization (VQ) method from VQ-VAE (Van Den Oord et al., 2017) and Discrete-Value Neural Communication (Liu et al., 2021) works to discretize multiview visual representations, $E_m \in (E_{ego}, E_{exo}, E_{top})$. Previous works use VQ to discretize a representation using codebooks, whereas we use shared codebooks to discretize and align multiview representations to learn unified concepts across visual views for extracting salient multimodal representations. First, VQ-Fusion divides each $E_m$ into $G$ continuous segments $(s_{(m,1)}, s_{(m,2)}, \ldots, s_{(m,G)})$, where $E_m = \text{CONCAT}(s_{(m,1)}, s_{(m,2)}, \ldots, s_{(m,G)})$ and

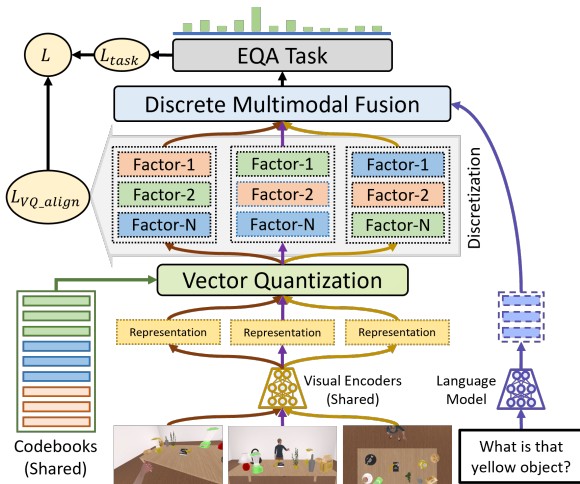

Figure 4: VQ-Fusion: Vector Quantization (VQ) based multimodal learning model architecture. VQ-Fusion extracts multiview visual representations using visual encoders, which are then discretized using shared codebooks. The shared codebooks' bottleneck allows the model to learn unified concepts across multiple views. Finally, discretized visual representations are fused with discrete verbal representations to produce multimodal representation.

$s_{(m,i)} \in \mathbb{R}^{D_m/G}$. Second, VQ-Fusion independently maps continuous segment $s_{(m,i)}$ to discrete latent code $c_j \in \mathbb{R}^{L \times (D_m/G)}$ using shared codebooks $C$, where $L$ is codebooks size (i.e., number of categorical codes in each codebook). We can find the optimal code for each continuous segment $s_{(m,i)}$ from the codebooks in the following way: $e_{(m,o_i)} = F^D(s_{(m,i)}), \quad o_i = \underset{j \in 1 \ldots L}{\arg\max} ||s_{(m,i)} - c_j||$ . Here, $F^D$ is the discretization ($D$) method. Finally, we concatenate the discretized codes to produce discretized visual representation $E_m^D$ in the following way: $E_m^D = \text{CONCAT}(F^D(s_{(m,1)}), \ldots, F^D(s_{(m,G)}))$ .

Following the training procedure in (Liu et al., 2021) and (Van Den Oord et al., 2017), we calculate VQ loss for learning the codebooks: $\mathcal{L}_{VQ\_align} = \frac{\beta}{G} \sum_i^G ||s_i - sg(c_{o_i})||_2^2$. Here, $sg$ is the stop-gradient operator blocking gradients to $c_{o_i}$, and $\beta$ is a hyperparameter controlling reluctance to change the code. We train the discretization module to learn codebooks using gradient descent with the other parts of VQ-Fusion. As VQ-Fusion employs shared codebooks to discretize visual representation for multiple views, $\mathcal{L}_{VQ\_align}$ loss aids in aligning multiview representations and learning unified concepts across views. This shared codebooks approach allows aligning multiview representation to answer the question with multimodal expressions effectively.

Finally, VQ-Fusion fuses these discretized visual and verbal representations using a self-attention approach to produce task representation: $E_{fused} = \sum_{m \in M} \alpha_m E_m$ . Here, $\alpha_m = \frac{exp(\gamma_m)}{\sum_{m \in M} exp(\gamma_m)}$ and $\gamma_m = (W)^T E_m, m \in M$ . Here, $M$ is the modality list (ego, exo, top, verbal), $W$ is a learnable parameter, and $\alpha_m$ is the attention score which is calculated using a 1D-CNN with a filter size of 1.

**Task Learning:** We use the fused representation, $E_{fused}$, to learn different EQA tasks $T_k$: $y_{T_k} = F_{T_k}(E_{fused})$. Here, $F_{T_k}$ is the task learning module, which can be designed based on the EQA task

Table 3: Comparisons of VL models performance for EQA tasks. The results suggest that incorporating VQ-Fusion in VL models can improve the performance of EQA tasks. ✓: VL models with VQ-Fusion, and ✗: VL models without VQ-Fusion.

| Models | EP | | OG | | POG | | OC | |
|---|---|---|---|---|---|---|---|---|
| | ✗ | ✓ | ✗ | ✓ | ✗ | ✓ | ✗ | ✓ |
| Dual Encoder | 53.46 | 55.78 | 48.31 | 49.96 | 83.91 | 84.28 | 12.28 | 12.38 |
| CLIP | 53.17 | 54.72 | 54.06 | 65.49 | 70.92 | 82.70 | 09.65 | 13.14 |
| VisualBERT | 50.00 | 54.51 | 53.39 | 54.50 | 86.09 | 87.09 | 14.09 | 14.35 |
| ViLT | 90.24 | 91.50 | 59.74 | 61.04 | 86.10 | 87.42 | 11.14 | 12.54 |

| Models | OAQ | | OAC | | PG | | RG | |
|---|---|---|---|---|---|---|---|---|
| | ✗ | ✓ | ✗ | ✓ | ✗ | ✓ | ✗ | ✓ |
| Dual Encoder | 63.71 | 66.90 | 57.92 | 61.45 | 66.72 | 66.77 | 75.78 | 89.36 |
| CLIP | 70.85 | 74.32 | 58.59 | 70.59 | 66.64 | 66.99 | 85.84 | 89.93 |
| VisualBERT | 51.43 | 54.45 | 58.56 | 59.98 | 66.37 | 79.11 | 89.13 | 89.26 |
| ViLT | 55.96 | 59.47 | 58.93 | 60.16 | 80.36 | 81.23 | 87.36 | 88.68 |

properties. For example, we use a multi-layer perceptron for the object existence task (Sect. 3 and please check the supplementary for further details). Moreover, $\mathcal{L}_{task,T_k}$ is used to train the model for task $T_k$: $\mathcal{L}_{task,T_k}(y_{T_k}, \hat{y}_{T_k}) = \frac{1}{B}\sum_{i=1}^{B} y_{(T_k,i)} \log \hat{y}_{(T_k,i)}$. Finally, we combine the task learning loss ($\mathcal{L}_{task,T_k}$) with the VQ loss ($\mathcal{L}_{VQ\_align}$) using task learning weights ($\mathcal{W}_{VQ}$ and $\mathcal{W}_{task}$) to train the VQ-Fusion model: $\mathcal{L} = \mathcal{W}_{VQ}\mathcal{L}_{VQ\_align} + \mathcal{W}_{task}\mathcal{L}_{task,T_k}$ .

**Variations of VQ-Fusion:** VQ-Fusion allows to use existing VL models (e.g., VisualBERT (Li et al., 2019) & ViLT (Kim et al., 2021)) to extract these representations. As the architecture of these VL transformer models is limited to processing a single visual and verbal input, we need to pair the verbal question to each visual view and pass through these models to extract multiview visual and verbal representations. We use these representations in VQ-Fusion to discretize and fuse to produce multimodal representations. Please check the supplementary materials for further details.

# 7 EXPERIMENTAL ANALYSIS

In this section, we have presented experimental analyses on our EQA-MX dataset to evaluate the impact of VQ-Fusion in VL models for EQA tasks. We have included additional ablation studies and experimental analyses for another task in the supplementary to evaluate the significance of VQ-Fusion for multimodal representation learning.

**Baseline Models:** Existing visual-language (VL) models for QA tasks are designed to answer a question using a single visual context. Since our proposed EQA tasks involve three visual views, we extend four VL models to learn multiview representations: Dual-Encoder (ViT+BERT) (Dosovitskiy et al., 2020; Devlin et al., 2018), CLIP (Radford et al., 2021), VisualBERT (Li et al., 2019), and ViLT (Kim et al., 2021). For the Dual-Encoder (ViT+BERT) model, we independently extract visual representations for each view using a shared ViT model and verbal representations using a BERT model. We fuse these visual and verbal representations to produce task representations. For the CLIP models, we pair each visual view to a verbal question and pass this through the model to extract multiple visual and verbal representations and fuse them to produce task representations. For VisualBERT and ViLT, we use ResNet-101 (He et al., 2016) to extract visual representations that are passed through the model with verbal embeddings to produce task representations. Please check the supplementary materials for further details.

## 7.1 COMPARISON OF MULTIMODAL LEARNING MODELS

We evaluated state-of-the-art visual-language (VL) models with and without our VQ-Fusion to learn VL representations for 8 EQA tasks. We varied the number of codebooks to $\{2, 4, 8, 16\}$ in VQ for each task and reported the best performance. We trained and evaluated these models independently for each task as a single-task model on our EQA-MX dataset. We used data samples with varying nonverbal gestures: gaze and pointing gestures, only gaze, and only pointing gestures. All the visual views (ego, exo, and top) and verbal perspectives (speaker, observer, and neutral) are used

to train models and evaluate whether the models can learn generalized representation from diverse data. We report macro-accuracy across all tasks to accurately gauge whether models can effectively understand EQA tasks and are not biased toward a particular class (Table 3).

**Results:** The results in Table 3 suggest that incorporating VQ-Fusion in VL models helps to successfully fuse extracted salient multiview representations with verbal representations, and thus improves model performance on EQA tasks. For example, the CLIP model without VQ-Fusion achieves $54.06\%$ accuracy in the object grounding task (OG), whereas incorporating VQ-Fusion in the CLIP model increases the OG task's performance to $65.49\%$. Similarly, VQ-Fusion improved the CLIP model's performance on the object attribute query task (OAQ) by $12\%$, the VisualBERT model's performance on the perspective grounding task by $12.74\%$, the ViLT model's performance on the object attribute comparison (OAC) task by $3.5\%$, and the DualEncoder model's performance on the relation grounding task (RG) by $13.58\%$. These performance improvements validate the significance of VQ-Fusion in extracting salient multimodal representations from multiple visual and verbal perspectives for effectively learning EQA tasks.

**Discussion:** The primary reasoning behind the performance improvement by incorporating VQ-Fusion in VL models lies in its discretization of multiview representations before fusion with discrete verbal representations. VQ-Fusion uses codebooks to discretize and align the visual representations with the discrete structure of verbal representations. Conversely, existing VL models extract continuous monolithic visual representations and fuse them with discrete verbal representations. This structural mismatch leads to sub-optimal multimodal fusion, adversely affecting the extraction of salient task representations and subsequently degrading task performance.

Moreover, as VQ-Fusion uses shared codebooks in the VQ information bottleneck to learn multimodal representations, this codebook sharing enables models to align the multiview representations and learn unified concepts. Learning unified concepts from multiple views is crucial, as multiple views capture the same interaction. Existing models are designed to learn visual and language representations from a single visual perspective. Thus, these models do not have any mechanisms to extract unified concepts from multiple visual views. VQ-Fusion enables these models to learn this unified concept using shared codebooks-based VQ.

Our experimental results also indicate that incorporating additional perspective-related information can help models to successfully ground objects. This is made apparent by the model performance on the perspective-aware object grounding (POG) task being consistently higher then the model performance on the object grounding (OG) task. This is particularly notable as the only difference between these tasks is the presence of the question's verbal perspective (Fig. 2). Thus, these results suggest models need to understand verbal perspective for successfully grounding objects in situations with multiple verbal perspectives.

Although the VL models presented can achieve considerable performance for most of the EQA tasks, these models perform slightly better than random-guessing for the object counting (OC) task. As these models do not use object location-specific information, the models suffer at locating and counting objects given a spatial relation. One possible extension of these models to improve performance for the OC task is learning mechanisms to push VL models to learn object locations. The EQA-MX dataset contains rich annotations of object locations, which can easily be incorporated in developing models more capable of understanding spatial locations.

## 7.2 Impact of Nonverbal Gestures (Ablation Study)

We evaluated the impact of nonverbal gestures on learning EQA tasks. We evaluated VQ-Fusion with CLIP models and $8$ codebooks on the different splits of EQA-MX dataset: data samples with gaze and gestures, only gaze, only gestures, and without gaze and gestures (this data split contains visual scenes without human).

**Results and Discussion:** The results in Table 4 suggest that the model performs is worse for EQA tasks if we train the model using data without nonverbal gestures. For example, the model trained using data without nonverbal gestures achieved only $26.65\%$ accuracy for the object grounding (OG) task, whereas the model trained using data with gaze and pointing gestures achieved $68.61\%$ accuracy for the OG task. This is a trend for all other tasks where the performance improved when gaze and/or pointing gestures were incorporated compared to when it only relied on the verbal

message. The performance degradation indicates that the models must learn nonverbal gestures to answer questions with multimodal expressions for EQA tasks.

## 7.3 IMPACT OF VQ CODEBOOKS (ABLATION STUDY)

We evaluated VQ-Fusion with the CLIP model for $8$ EQA tasks by varying the number of codebooks in VQ: $\{2, 4, 8, 16\}$. We evaluated these models on our EQA-MX with varied nonverbal gestures (gaze and pointing gestures, only gaze, and only pointing gestures). We trained these models with multiple visual and verbal perspectives.

**Results and Discussion:** The results in Table 5 suggest that different codebooks help the model achieve the highest performance for different tasks. For example, VQ-Fusion with $8$ codebooks can achieve the highest performance in existence prediction (EP), object grounding (OG), and object attribute compare (OAC) tasks, whereas VQ-Fusion with $2$ codebooks can achieve the highest

Table 4: Impact of gaze (G) and pointing gestures (PG) in learning EQA tasks. The results suggest that incorporating gestures improves EQA task performance. G (✗) and PG (✗) indicate visual scenes that do not include humans.

| G | PG | EQA Tasks | | | | | | | |
|---|---|---|---|---|---|---|---|---|---|
| | | EP | OG | POG | OC | OAQ | OAC | PG | RG |
| ✗ | ✗ | 51.03 | 26.65 | 52.79 | 09.94 | 24.01 | 51.22 | 48.95 | 56.75 |
| ✗ | ✓ | 53.87 | 60.66 | 71.08 | 11.51 | 64.69 | 60.63 | 66.31 | **90.01** |
| ✓ | ✗ | 53.51 | 63.49 | 70.90 | **12.29** | 69.43 | **61.25** | 66.67 | 87.23 |
| ✓ | ✓ | **54.38** | **68.61** | **79.68** | 11.86 | **72.62** | 60.74 | **66.68** | 89.59 |

performance for perspective-aware object grounding (POG) and object counting (OC) tasks. The number of codebooks depends on the task complexity of how many concepts need to be learned. As the OG task requires learning verbal perspective, the model requires more codebooks to learn perspective-related concepts. On the other hand, as perspective is already given in the POG task, VQ-Fusion requires fewer codebooks. Our results also show similar phenomena, where VQ-Fusion achieves $82.70\%$ accuracy for the POG task with only $2$ codebooks, whereas it achieves $65.49\%$ accuracy for the OG task with $8$ codebooks.

However, increasing codebooks more than optimal leads to decreasing task performance. For example, the object attributes compare (OAC) task accuracy degrades if we increase the number of codebooks to more than $4$. As the OAC task involves whether two objects have the same attribute, the model can learn these simple concepts using fewer codebooks. In-

Table 5: Impact of the number of VQ codebooks (VQ CBs) in VQ-Fusion with the CLIP model in learning EQA tasks.

| VQ CBs | EQA Tasks | | | | | | | |
|---|---|---|---|---|---|---|---|---|
| | EP | OG | POG | OC | OAQ | OAC | PG | RG |
| 2 | 53.46 | 64.86 | **82.70** | **13.14** | 61.39 | 57.43 | 61.39 | 88.24 |
| 4 | 52.15 | 61.12 | 73.94 | 11.35 | 69.42 | **70.59** | 60.30 | **89.93** |
| 8 | **54.72** | **65.49** | 73.97 | 11.92 | **70.85** | 60.68 | 66.82 | 88.23 |
| 16 | 53.19 | 55.12 | 71.32 | 11.43 | 69.35 | 60.37 | **66.99** | 84.36 |

creasing the number may lead to sparsity in codebooks, i.e., many codes are left unutilized, hindering models from extracting salient representations. On the other hand, using a few codebooks for complex tasks, such as OG and OAQ, leads to tight bottlenecks, which deters models from learning salient concepts. This results in lower task performance. These results indicate that each task has a different optimal number of codebooks.

## 8 CONCLUSION

To develop models for comprehending embodied interactions, we designed $8$ novel EQA tasks requiring comprehension of questions with multimodal expressions (verbal and nonverbal gestures). To train and diagnose models for these EQA tasks, we developed a novel large-scale dataset, EQA-MX, which contains questions with multimodal expressions from multiple verbal and visual perspectives. Moreover, we developed a vector quantization-based multimodal representation learning model, VQ-Fusion, to learn salient multimodal representation from multiple visual and verbal perspectives. Our extensive experimental analyses suggest that VQ-Fusion can effectively fuse continuous multiview visual and discrete verbal representation, which helps to improve the visual-language model's performance for all EQA tasks up to $13\%$.

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

TECHNICAL APPENDIX

# A  RESOURCES

The EQA-MX dataset, source code for the CAESAR simulator with our modifications, benchmark learning models, trained model checkpoints, and docker for computing environment can be accessed through the following links. We will publicly release these resources with the camera-ready version of our paper. For double-blind reviewing purposes, we are sharing these resources anonymously with the reviewers:

- **EQA-MX dataset** (162 **GB**): `https://bit.ly/eqa-mx-dataset`
- **Source code of VQ-Fusion, benchmark models, dataset processing, and dataset analyses:**
  `https://bit.ly/eqa-repo`
- **Source code of the EQA simulator with new extensions** (3.1 **GB**): `https://bit.ly/eqa-simulator`
- **Trained model checkpoints of ViLT with VQ-Fusion for existence prediction task** (1.3 **GB**):
  `https://bit.ly/model-checkpoint`
- **Docker for training models** (8.59 **GB**): We built a docker to facilitate easy reproducing of our experimental settings and training environment. We cannot currently share the docker hub link to maintain anonymity. We plan to share that docker link upon publication of the paper. For this reason, we are sharing the singularity container built from the same docker we used for our experimentation: `https://bit.ly/multimodal-docker`

# B  BROADER IMPACT

Our dataset contains rich annotations of visual scenes, such as object locations, spatial relations, and multiple visual and verbal perspectives. These can be used to design new tasks to robustly comprehend embodied interactions. Moreover, our EQA-MX dataset can be used for diverse tasks in embodied settings, such as scene segmentation and conversational human-AI interactions with multimodal expressions. Additionally, our dataset can be used to develop and evaluate models that can be transferred to robots for comprehending embodied human instructions in real-world settings. Lastly, our experimental analysis provides valuable insights that can be used in designing robust VL models, such as using similar embedding structures for fusing continuous and discrete representations leading to performance improvements.

# C  ADDITIONAL EXPERIMENTAL ANALYSES

## C.1  IMPACT OF MULTIPLE VISUAL PERSPECTIVES AND MODALITIES

In real-world settings, robots are typically equipped with multiple camera views. Several studies have emphasized the significance of multiview data in accurately comprehending human actions and instructions(Kong et al., 2019; Islam & Iqbal, 2022). To further validate the importance of multimodal data (nonverbal gestures captured through visual views and verbal utterances) in understanding embodied question answering (EQA) tasks, we conducted extensive ablation studies with varying visual views (ego, exo and top) and verbal utterances (verbal utterance templates described in Table 11).

In the first setting, we used only verbal utterances for all eight EQA tasks (Table,6: Top). We used BERT (Devlin et al., 2018) for learning the EQA tasks. The results suggest models using only a verbal modality can not effectively learn these EQA tasks. Conversely, if we utilized both verbal and nonverbal data, then the performance of these EQA tasks improved (Table,6). This degraded performance using only verbal data emphasizes the importance of utilizing both verbal and nonverbal data modalities for appropriately learning EQA tasks. Additionally, it also indicates that our proposed EQA-MX dataset is less biased towards verbal data for comprehending EQA tasks.

Table 6: We trained CLIP models with VQ-Fusion using different combinations of modalities on the 8 tasks described in Figure 2 in the paper. Top Table: only verbal questions. Bottom Table: different visual modalities and verbal questions. The results suggest that multimodal models outperform those using only verbal data (Top Table). Additionally, training models with multiview data leads to robust performance, while using a subset of views results in performance degradation if the views change during testing (Bottom Table). Existence Prediction (EP), Object Grounding (OG), Perspective-Aware Object Grounding (POG), Object Counting (OC), Object Attribute Query (OAQ), Object Attribute Compare (OAC), Perspective Grounding (PG), Relation Grounding (RG).

| | | EP | OG | POG | OC | OAQ | OAC | PG | RG |
|---|---|---|---|---|---|---|---|---|---|
| Only Verbal | | 40.64 | 8.90 | 45.46 | 7.45 | 7.69 | 29.49 | 45.23 | 44.82 |
| Train | Test | EP | OG | POG | OC | OAQ | OAC | PG | RG |
| Ego | Ego | 53.86 | 59.92 | 70.98 | 10.60 | 68.56 | 61.86 | 64.41 | 87.54 |
| Ego | Exo | 52.61 | 17.28 | 62.45 | 8.96 | 15.06 | 56.62 | 63.39 | 82.33 |
| Exo | Exo | 53.67 | 39.46 | 69.96 | 11.24 | 56.76 | 60.20 | 66.39 | 88.58 |
| Exo | Ego | 52.84 | 21.39 | 69.70 | 10.78 | 25.03 | 58.68 | 64.49 | 88.20 |
| ALL | ALL | 54.72 | 65.49 | 82.70 | 13.14 | 74.32 | 70.59 | 66.99 | 89.93 |
| ALL | Ego | 54.32 | 60.63 | 82.31 | 12.22 | 69.84 | 60.89 | 66.71 | 89.03 |
| ALL | Exo | 54.17 | 59.14 | 78.02 | 12.55 | 61.71 | 62.25 | 66.53 | 89.26 |

In the second setting, we used verbal utterances and nonverbal gestures to learn EQA tasks. We varied the visual perspectives during training and testing through the use of different camera views (ego, exo, and top) to capture the nonverbal interactions. We used CLIP model to learn EQA tasks involving verbal utterances and visual views. The results suggest that models trained using multiple visual perspectives perform better than models trained using a single visual perspective (Table,6: Bottom). The reasoning behind this performance improvement is that models using multiple visual views can learn generalized multiview representations, which can improve the performance at inference time when visual views are varied.

## C.2 COMPARISON OF SINGLE AND MULTITASK MODELS

We evaluated the impact of learning multiple tasks in a visual-language model. We conducted this experimental analysis in two settings. In both settings, we used verbal utterances and multiple visual modalities to learn EQA tasks. In the first setting, we trained CLIP models for each EQA task separately. In the second setting, we trained CLIP models for a subset of EQA tasks. Finally, we used the extracted representation in each EQA task head, where these task heads are designed using an MLP.

The results in Table 7 suggest that the performance of models learning multiple tasks degrades compared to the models learning these tasks separately. As these tasks have different characteristics, learning these tasks together can compete in the representation learning space and degrades these tasks' performance. For example, training the CLIP model for the Existence Prediction (EP) and Object Grounding (OG) tasks together degrades the Object Grounding task performance to 40.76% compared to an accuracy of 65.49% for a separately trained CLIP model for OG task. Previous studies have observed similar performance degradation when learning multiple competing tasks. The primary reason behind the performance degradation is that the competing tasks have conflicting gradients among different tasks that introduce negative knowledge transfer and thus degrade these tasks' performance. Thus, an exciting future research direction would be to design novel multi-task model architectures and training approaches where training on multiple tasks using multiple modalities improves the performance of every task in a shared model.

## C.3 GENERALIZABILITY OF VQ-FUSION

To evaluate the generalizability of VQ-Fusion for another task involving multimodal representation learning, we incorporate VQ-Fusion in an existing multimodal learning model (HAMLET (Islam & Iqbal, 2020)) for human activity recognition tasks with multimodal sensor data (RGB videos, acceleration, gyroscope, and orientation). We have evaluated this modal on the MMAct dataset (Kong

Table 7: We train CLIP models with VQ-Fusion in single task (ST) and multitask (MT) settings. We reported accuracy of these tasks. Tasks trained in an MT setting are grouped together. The results suggest that the performance of these models with multiple tasks degrades compared to models learning these tasks separately. Existence Prediction (EP), Object Grounding (OG), Perspective-Aware Object Grounding (POG), Object Counting (OC), Object Attribute Query (OAQ), Object Attribute Compare (OAC), Perspective Grounding (PG), Relation Grounding (RG).

| ST | EP | OG | POG | OC | OAQ | OAC | PG | RG |
|---|---|---|---|---|---|---|---|---|
|  | 54.72 | 65.49 | 82.70 | 13.14 | 74.32 | 70.59 | 66.99 | 89.93 |

| MT | EP | OG | | EP | POG | | EP | PG |
|---|---|---|---|---|---|---|---|---|
|  | 53.25 | 40.76 | | 52.68 | 73.90 | | 52.62 | 49.86 |

| MT | EP | OAQ | OG | | EP | PG | OAQ | | PG | EQ | OAQ |
|---|---|---|---|---|---|---|---|---|---|---|---|
|  | 54.24 | 68.70 | 55.56 | | 53.17 | 66.92 | 66.61 | | 66.80 | 53.26 | 69.01 |

et al., 2019). The MMAct dataset comprises 37 common daily life activities, each performed by 20 individuals and repeated five times. The dataset includes seven modalities, ranging from RGB data to acceleration and gyroscope measurements. Our experiments focused on utilizing two available viewpoints of RGB videos, as well as acceleration, gyroscope, and orientation data. Notably, the MMAct dataset also includes visually occluded data samples, providing an opportunity to evaluate the effectiveness of multimodal learning approaches in extracting complementary features for activity recognition.

In our experimental analyses, we adhered to the original session-based evaluation settings and reported the F1-score. We have used eight codebooks to discretize the multimodal representations. The results indicated that the HAMLET model, which utilizes our proposed VQ-Fusion approach, outperformed all existing state-of-the-art multimodal human activity recognition (HAR) approaches in session-based evaluation settings on the MMAct dataset (Table 8). Specifically, the inclusion of VQ-Fusion enabled HAMLET to improve its F1-score by $4.2\%$, resulting in the highest reported F1-score of $87.69\%$ (Table 8). These findings suggest that VQ-Fusion can effectively aid existing models in extracting salient multimodal representations, thereby enhancing the performance of downstream tasks in the field of HAR.

## D  TRAINING ENVIRONMENT

We developed all the models using the Pytorch (version: 1.12.1+cu113) (Paszke et al., 2019) and Pytorch-Lightning (version: 1.7.1) (Falcon, 2019) deep learning frameworks. We also used Hug-

Table 8: Cross-session performance comparison (F1-Score) of multimodal learning methods on MMAct dataset

| Method | F1-Score (%) |
|---|---|
| SVM+HOG (Ofli et al., 2013) | 46.52 |
| TSN (RGB) (Wang et al., 2016) | 69.20 |
| TSN (Optical-Flow) (Wang et al., 2016) | 72.57 |
| MMAD (Kong et al., 2019) | 74.58 |
| TSN (Fusion) (Wang et al., 2016) | 77.09 |
| MMAD (Fusion) (Kong et al., 2019) | 78.82 |
| Keyless (Long et al., 2018) | 81.11 |
| HAMLET (Islam & Iqbal, 2020) | 83.89 |
| MuMu (Islam & Iqbal, 2022) | 87.50 |
| VQ-Fusion(HAMLET) | **87.69** |
| VQ-Fusion(MuMu) | **87.83** |

Table 9: Comparison of the QA datasets. Existing VQA and EQA datasets do not contain nonverbal human gestures (NV), multiple verbal perspectives (MV), contrastive (C) and ambiguous (A) data samples. ‡ Embodied (E) interactions refer to humans interacting with multimodal expressions. † Embodied interactions refer to an agent navigating in an environment. V: Verbal and MT: Multitasks.

| Datasets | V | NV | E | EQA | MT | MV | Views Exo | Ego | Top | C | A |
|---|---|---|---|---|---|---|---|---|---|---|---|
| PointAt (Schauerte et al., 2010) | ✗ | ✓ | ✓ | ✗ | ✗ | ✗ | ✓ | ✗ | ✗ | ✗ | ✗ |
| ReferAt (Schauerte & Fink, 2010) | ✓ | ✓ | ✓ | ✗ | ✗ | ✗ | ✓ | ✗ | ✗ | ✗ | ✗ |
| IPO (Shukla et al., 2015) | ✗ | ✓ | ✓ | ✗ | ✗ | ✗ | ✓ | ✗ | ✗ | ✗ | ✗ |
| IMHF (Shukla et al., 2016) | ✗ | ✓ | ✓ | ✗ | ✗ | ✗ | ✓ | ✗ | ✗ | ✗ | ✗ |
| RefIt (Kazemzadeh et al., 2014) | ✓ | ✗ | ✗ | ✗ | ✗ | ✗ | ✓ | ✗ | ✗ | ✗ | ✗ |
| RefCOCO (Yu et al., 2016) | ✓ | ✗ | ✗ | ✗ | ✗ | ✗ | ✓ | ✗ | ✗ | ✗ | ✗ |
| RefCOCO+ (Yu et al., 2016) | ✓ | ✗ | ✗ | ✗ | ✗ | ✗ | ✓ | ✗ | ✗ | ✗ | ✗ |
| RefCOCOg (Mao et al., 2016) | ✓ | ✗ | ✗ | ✗ | ✗ | ✗ | ✓ | ✗ | ✗ | ✗ | ✗ |
| Flickr30k (Plummer et al., 2015) | ✓ | ✗ | ✗ | ✗ | ✗ | ✗ | ✓ | ✗ | ✗ | ✗ | ✗ |
| GuessWhat? (De Vries et al., 2017) | ✓ | ✗ | ✗ | ✗ | ✗ | ✗ | ✓ | ✗ | ✗ | ✗ | ✗ |
| Cops-Ref (Chen et al., 2020b) | ✓ | ✗ | ✗ | ✗ | ✗ | ✗ | ✓ | ✗ | ✗ | ✗ | ✗ |
| CLEVR-Ref+ (Liu et al., 2019) | ✓ | ✗ | ✗ | ✗ | ✓ | ✗ | ✓ | ✗ | ✗ | ✗ | ✗ |
| DAQUAR (Malinowski et al., 2017) | ✓ | ✗ | ✗ | ✗ | ✗ | ✗ | ✓ | ✗ | ✗ | ✗ | ✗ |
| FM-IQA (Gao et al., 2015) | ✓ | ✗ | ✗ | ✗ | ✗ | ✗ | ✓ | ✗ | ✗ | ✗ | ✗ |
| Visual Madlibs (Yu et al., 2015) | ✓ | ✗ | ✗ | ✗ | ✓ | ✗ | ✓ | ✗ | ✗ | ✗ | ✗ |
| Visual Genome (Krishna et al., 2017) | ✓ | ✗ | ✗ | ✗ | ✓ | ✗ | ✓ | ✗ | ✗ | ✗ | ✗ |
| DVQA (Kafle et al., 2018) | ✓ | ✗ | ✗ | ✗ | ✓ | ✗ | ✓ | ✗ | ✗ | ✗ | ✗ |
| VQA (COCO) (Antol et al., 2015) | ✓ | ✗ | ✗ | ✗ | ✓ | ✗ | ✓ | ✗ | ✗ | ✗ | ✗ |
| VQA (Abs.) (Antol et al., 2015) | ✓ | ✗ | ✗ | ✗ | ✓ | ✗ | ✓ | ✗ | ✗ | ✗ | ✗ |
| Visual 7W (Zhu et al., 2016) | ✓ | ✗ | ✗ | ✗ | ✓ | ✗ | ✓ | ✗ | ✗ | ✗ | ✗ |
| KB-VQA (Wang et al., 2015) | ✓ | ✗ | ✗ | ✗ | ✓ | ✗ | ✓ | ✗ | ✗ | ✗ | ✗ |
| FBQA (Wang et al., 2017) | ✓ | ✗ | ✗ | ✗ | ✓ | ✗ | ✓ | ✗ | ✗ | ✗ | ✗ |
| VQA-MED (Hasan et al., 2018) | ✓ | ✗ | ✗ | ✗ | ✗ | ✗ | ✓ | ✗ | ✗ | ✗ | ✗ |
| DocVQA (Mathew et al., 2021) | ✓ | ✗ | ✗ | ✗ | ✓ | ✗ | ✓ | ✗ | ✗ | ✗ | ✗ |
| YouRefIt (Chen et al., 2021) | ✓ | ✓ | ✓ | ✗ | ✓ | ✗ | ✓ | ✗ | ✗ | ✗ | ✗ |
| GRiD-3D (Lee et al., 2022) | ✓ | ✗ | ✗ | ✗ | ✓ | ✗ | ✓ | ✗ | ✗ | ✗ | ✗ |
| EQA † (Das et al., 2018a) | ✓ | ✗ | ✗ | ✗ | ✓ | ✗ | ✓ | ✗ | ✗ | ✗ | ✗ |
| MT-EQA † (Das et al., 2018a) | ✓ | ✗ | ✗ | ✗ | ✓ | ✗ | ✓ | ✗ | ✗ | ✗ | ✗ |
| CAESAR-L (Islam et al., 2022b) | ✓ | ✓ | ✓ | ✗ | ✗ | ✓ | ✓ | ✓ | ✓ | ✓ | ✓ |
| CAESAR-XL (Islam et al., 2022b) | ✓ | ✓ | ✓ | ✗ | ✗ | ✓ | ✓ | ✓ | ✓ | ✓ | ✓ |
| EQA-MX ‡ | ✓ | ✓ | ✓ | ✓ | ✓ | ✓ | ✓ | ✓ | ✓ | ✓ | ✓ |

gingFace library (version: 4.21.1) for pre-trained models (BERT [1] (Devlin et al., 2018), ViT [2] (Dosovitskiy et al., 2020), VisualBERT [3] (Li et al., 2019), Dual Encoder [4], ViLT [5](Kim et al., 2021), and CLIP [6] (Radford et al., 2021)). For the Dual-Encoder and CLIP models, we used an embedding size of 512, and for VisualBERT and ViLT, we used an embedding size of 768. We train models using the Adam optimizer with a weight decay regularization (Loshchilov & Hutter, 2017) and cosine annealing warm restarts at an initial learning rate: $3e^{-4}$, cycle length ($T_0$): 4, and cycle multiplier ($T_{mult}$): 2. We used batch size 128 and trained models for 8 epochs. We used the same fixed random seed (33) for all the experiments to ensure reproducibility. Lastly, all models are trained in distributed GPU clusters, where each node contains 8 A100 GPUs.

---

[1] https://huggingface.co/docs/transformers/model_doc/bert

[2] https://huggingface.co/docs/transformers/model_doc/vit

[3] https://huggingface.co/docs/transformers/model_doc/visual_bert

[4] https://huggingface.co/docs/transformers/model_doc/vision-text-dual-encoder

[5] https://huggingface.co/docs/transformers/model_doc/vilt

[6] https://huggingface.co/docs/transformers/model_doc/clip

| Datasets | No. of Images | No. of Samples | Object Categories | Avg. Words* |
|---|---|---|---|---|
| PointAt (Schauerte et al., 2010) | 220 | 220 | 28 | - |
| ReferAt (Schauerte & Fink, 2010) | 242 | 242 | 28 | - |
| IPO (Shukla et al., 2015) | 278 | 278 | 10 | - |
| IMHF (Shukla et al., 2016) | 1716 | 1716 | 28 | - |
| RefIt (Kazemzadeh et al., 2014) | 19,894 | 130,525 | 238 | 3.61 |
| RefCOCO (Yu et al., 2016) | 19,994 | 142,209 | 80 | 3.61 |
| RefCOCO+ (Yu et al., 2016) | 19,992 | 141,564 | 80 | 3.53 |
| RefCOCOg (Mao et al., 2016) | 26,711 | 104,560 | 80 | 8.43 |
| Flickr30k (Plummer et al., 2015) | 31,783 | 158,280 | 44,518 | - |
| GuessWhat? (De Vries et al., 2017) | 66,537 | 155,280 | - | - |
| Cops-Ref (Chen et al., 2020b) | 75,299 | 148,712 | 508 | 14.40 |
| CLEVR-Ref+ (Liu et al., 2019) | 99,992 | 998,743 | 3 | 22.40 |
| DAQUAR (Malinowski et al., 2017) | 1449 | 124,68 | 37 | 11.5 |
| FM-IQA (Gao et al., 2015) | 157,392 | 316,193 | - | 7.38 |
| Visual Madlibs (Yu et al., 2015) | 107,38 | 360,001 | - | 6.9 |
| Visual Genome (Krishna et al., 2017) | 108,000 | 1,445,332 | 37 | 5.7 |
| DVQA (Kafle et al., 2018) | 300,000 | 3,487,194 | - | - |
| VQA (COCO) (Antol et al., 2015) | 204,721 | 614,163 | 80 | 6.2 |
| VQA (Abs.) (Antol et al., 2015) | 50,000 | 150,000 | 100 | 6.2 |
| Visual 7W (Zhu et al., 2016) | 47,300 | 327,939 | 36,579 | 6.9 |
| KB-VQA (Wang et al., 2015) | 700 | 5826 | 23 | 6.8 |
| FBQA (Wang et al., 2017) | 2190 | 5826 | 32 | 9.5 |
| VQA-MED (Hasan et al., 2018) | 2866 | 6413 | - | - |
| DocVQA (Mathew et al., 2021) | 12,767 | 50,000 | - | - |
| YouRefIt (Chen et al., 2021) | 497,348 | 4,195 | 395 | 3.73 |
| GRiD-3D (Lee et al., 2022) | 8,000 | 445,000 | 28 | - |
| EQA [†] (Das et al., 2018a) | 5,000 | 5,000 | 50 | - |
| MT-EQA [†] (Das et al., 2018a) | 19,287 | 19,287 | 61 | - |
| CAESAR-L (Islam et al., 2022b) | 11,617,626 | 124,412 | 61 | 5.56 |
| CAESAR-XL (Islam et al., 2022b) | 841,620 | 1,367,305 | 80 | 5.32 |
| EQA-MX [‡] | 750,849 | 8,243,893 | 52 | 11.45 |

Table 10: Comparison of the QA datasets. EQA-MX has more samples than all other previous QA datasets. *Average number of words in questions.

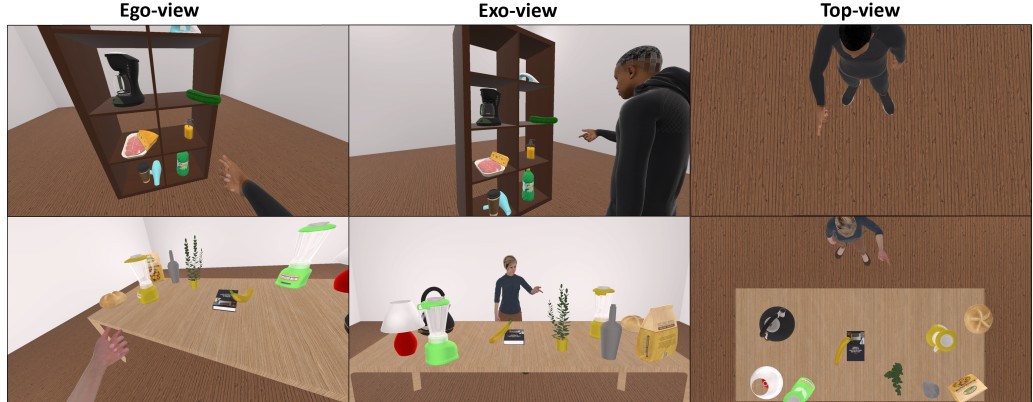

Figure 5: Sample data demonstrating the shelf environment vs. the table environment

Table 11: Templates for all 8 tasks in the EQA-MX dataset. The answers for these templates are based on the environment in the first row of 5.

| Existence Prediction | |
| --- | --- |
| Template | Task Example |
| Is there any/a/an <object name> in the scene? | Is there a cucumber in the scene? |
| Is there any/a/an <object color> <object name> in the scene? | Is there a green cucumber in the scene? |

| Object Grounding | |
| --- | --- |
| Template | Task Example |
| What is the name of that object/thing? | What is the name of that object? |
| What is the name of the <object color> object/thing? | What is the name of that yellow thing? |
| What is the name of that <object absolute location> <object color> object/thing? | What is the name of that right yellow object? |
| What is the name of that <selected object absolute location> <selected object color> object/thing to the <spatial relation> of the <relational object absolute location> <relational object color> <relational object name>? | What is the name of that right yellow object to the right of the yellow cheese? |

| Perspective-Aware Object Grounding | |
| --- | --- |
| Template | Task Example |
| Considering the <observer's/speaker's> perspective, what is the name of that object/thing? | Considering the observer's perspective, what is the name of that object? |
| Considering the <observer's/speaker's> perspective, what is the name of the <object color> object/thing? | Considering the observer's perspective, what is the name of that yellow thing? |
| Considering the <observer's/speaker's> perspective, what is the name of that <object absolute location> <object color> object/thing? | Considering the speaker's perspective, what is the name of that right yellow object? |

| Object Counting | |
| --- | --- |
| Template | Task Example |
| How many objects are <spatial relation> of the object/thing? | How many objects are above the object? |
| How many objects are <spatial relation> of the <object color> object/thing? | How many objects are left of the yellow thing? |

| Object Attribute Query | |
| --- | --- |
| Template | Task Example |
| What it the color of that object/thing? | What it the color of that object/thing? |
| What is the color of the <object name>? | What is the color of the hand soap dispenser? |

| Object Attribute Compare | |
| --- | --- |
| Template | Task Example |
| Is the color of that object/thing the same color as the <relational object name>? | Is the color of that thing the same color as the cheese? |
| Is the color of that <selected object name> the same color as the <relational object name>? | Is the color of that hand soap dispenser the same color as the soda bottle? |

| Perspective Grounding | |
| --- | --- |
| Template | Task Example |
| <Referring expressions using the templates from CAESAR>. From which perspective is the object described? | The hand soap dispenser above the soda bottle. From which perspective is the object described? |

| Relation Grounding | |
| --- | --- |
| Template | Task Example |
| <Referring expressions using the templates from CAESAR>, is the object referred to appropriately? | The hand soap dispenser above the cucumber, is the object referred to appropriately? |
| Considering the observer's perspective, <Referring expressions using the templates from CAESAR>, is the object referred to appropriately? | Considering the observer's perspective, the hand soap dispenser below the cucumber, is the object referred to appropriately? |
| Considering the speaker's perspective, <Referring expressions using the templates from CAESAR>, is the object referred to appropriately? | Considering the observer's perspective, the hand soap next to the coffee maker, is the object referred to appropriately? |

# E  EMBODIED QUESTION ANSWERING TASK AND DATASET ADDITIONAL INFORMATION

We include additional information on the EQA-MX dataset compared to previous EQA datasets in Tables 9 and 10.

All 8 EQA tasks we created described in the paper are catered towards specific scenarios that would benefit models in real-world human interaction scenarios.

**Existence Prediction (EP):** Naturally, humans are able to determine what objects are present in a given scene. In scenarios where humans are interacting and an actor mistakenly references an object not in the scene, this allows observers to request more information. Created to mimic this situation, the existence prediction task involves determining whether the scene contains a particular object with some specific attributes, such as color.

**Object Grounding (OG):** Understanding which objects a human refers to using verbal and non-verbal cues is key in successful human-AI interaction. A model successfully able to ground objects has use-cases such as assisting surgeons during a procedure by handing surgeons the correct tools. Thus, we design the object grounding task around this scenario, where models must identify the name of the object being referred to by verbal and nonverbal expressions.

**Perspective-Aware Object Grounding (POG):** Similar to the object grounding task, the Perspective-Aware Object Grounding involves determining which object is being referred to, but this task includes the verbal perspective (either ego, exo, or neutral). Although real-world human-AI interactions will not always contain the perspective of a given relation, including the perspective allows for us to determine whether or not understanding perspective can help in grounding objects.

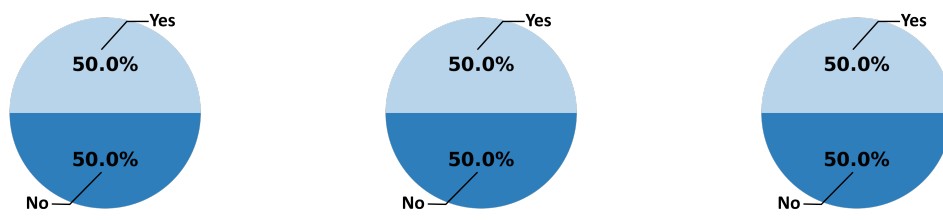

(a) Existence Prediction Task     (b) Object Attribute Compare Task     (b) Relation Grounding Task

Figure 6: Distributions of task outputs in the existence prediction (EP), object attribute compare (OAC), and relation grounding (RG) tasks. All these tasks have balanced binary outputs

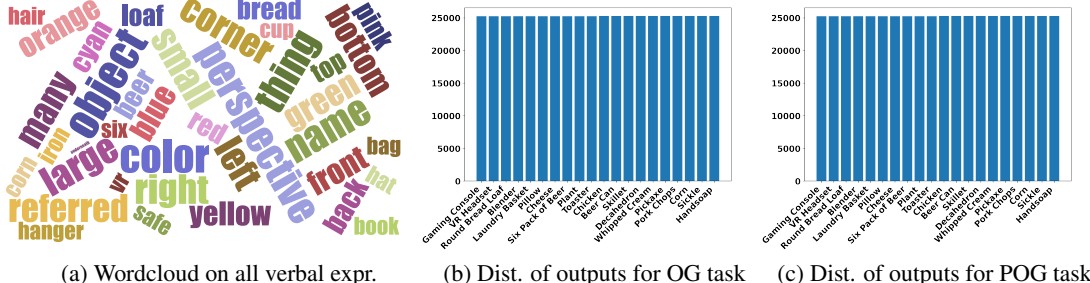

(a) Wordcloud on all verbal expr.     (b) Dist. of outputs for OG task     (c) Dist. of outputs for POG task

Figure 7: A verbal expression Wordcloud for the EQA-MX dataset, as well as the output distribution for the object grounding (OG) and perspective-aware object grounding (POG) tasks. In the Wordcloud the size of words represents the frequencies that they occur in the verbal utterances. Therefore, the most frequent words describe general properties of objects or are general words inside questions - such as color, perspective, and spatial relations/locations. In the diagrams for object frequencies for the object grounding and perspective-aware object grounding tasks, the most referred objects all have the same frequencies (these tasks have the same object distributions). Expr. : expression, Dist. : distribution.

**Object Counting (OC):** As understanding what object a human is referring to in a scene involves interpretation of the different number of objects inside that scene, understanding the number of objects in a scene can serve as an auxiliary task for the object grounding task. If models are able

to create salient multimodal representations to attend to all the objects in a given scene, is is likely they will be able to ground particular objects better. Thus, in the object counting task the number of objects in a scene is asked based on different spatial relations.

**Object Attribute Query (OAQ):** It is often important in human-human interactions to identify particular attributes of objects. Additionally, this information can be used as auxiliary information for tasks such as the Object Grounding task, where the goal is to identify objects. We design the Object Attribute Query task around this particular situation, where the color of a given object is queried for.

**Object Attribute Compare (OAC):** Humans often exchange information throughout conversations through the use of comparison of different object attributes. This exchange of information can assist in understanding the different objects an actor is referring to. Thus, we design the object attribute compare task, where the attributes of two different objects in the scene are compared.

**Perspective Grounding (PG):** Understanding human verbal perspective is integral to successful human-AI communication, as humans interchangeably describe objects from their perspective as well as the perspective of others. We simulate this in the perspective grounding task using three different perspectives - neutral, egocentric (speaker), and exocentric (observer).

**Relation Grounding (RG):** As described in Islam et al. (2022b), the relation grounding task involves determining whether the supplied verbal and nonverbal signals align with respect to describing the same object. Understanding whether or not a human is accurately verbally and nonverbally referring to an object can enable the identification of human mistakes. We add complexity to this task through the variation of verbal perspective in the question.

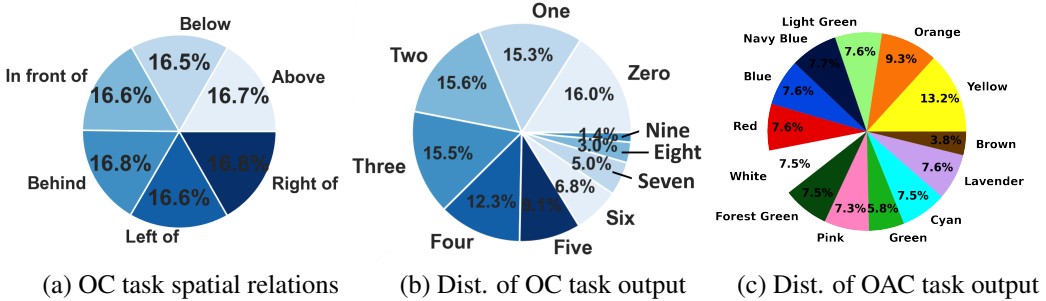

    (a) OC task spatial relations      (b) Dist. of OC task output      (c) Dist. of OAC task output

Figure 8: Distribution of task outputs in the object counting and object attribute compare tasks. Both distributions are not completely even due to different observed scene probabilities. For the object counting (OC) task, lower numbers have higher probabilities of occurring due to the number of objects in the scene ranging from $4 - 10$, hence the imbalance in distributions. Similarly, in the object attribute compare task different object colors are queried for, and since the colors of objects is not completely balanced, the task distribution is imbalanced. Dist. = Distribution.

### E.1 EQA TASK TEMPLATES

In this work we presented 8 EQA tasks. Each of these tasks has multiple sub-templates, which we present in more detail in Table 11. Each sub-template has multiple degrees of freedom from which to vary, ensuring generated embodied questions are diverse. For example, since most sub-templates use the absolute location of an object, this absolute location can often times be described from either the observer or speaker perspective.

### E.2 NEW ENVIRONMENTS IN EQA-MX

To increase dataset generalizability, we have added a shelf environment into the CAESAR simulator, and thus into the EQA-MX dataset. We visualize the three views (ego, exo, and top) for this and the table environment in Fig. 5. Because the exo and ego views in the table environment are on different sides of the table, the verbal perspectives differ. However, in the shelf environment, the exo and ego views are aligned meaning the verbal perspective is aligned. We created this environment in this way to ensure models have differing situations with regards to views and perspective. Additionally,

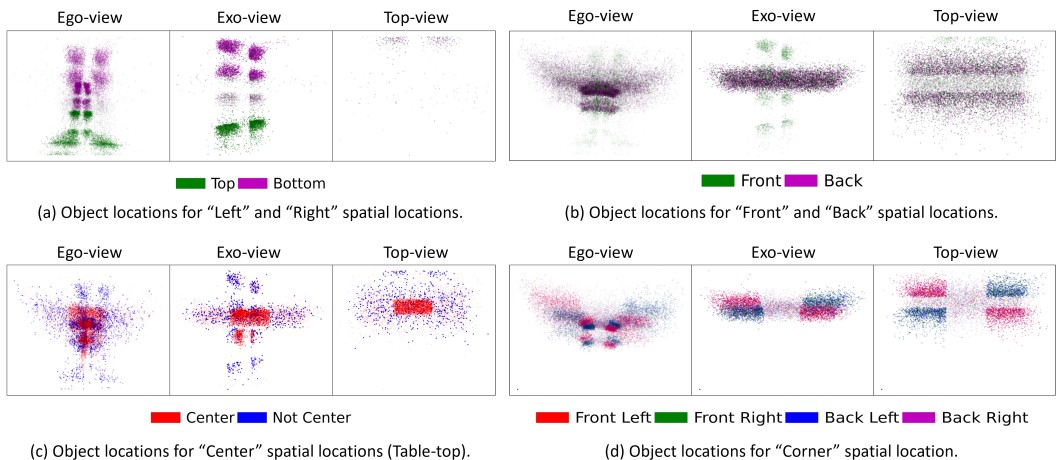

Figure 9: Object locations visualized for different spatial relations/locations across the EQA-MX dataset. The object locations are not easily separable based on spatial relations/locations that vary based on perspectives. (a & b) demonstrates how the shelf environment has more non-separable locations/relations due to the fact that verbal perspective in the shelf environment does not vary based on visual perspective. c is generally linearly separable, as expected, as the center of a given scene is objective. d demonstrates how opposing corners (i.e. front left and back right) are non-separable due to varying based on verbal perspectives).

since the shelf has objects below/on top of one another, it adds diversity with respect to spatial relations/locations, ensuring models understanding these relations/locations in all 3 dimensions.

## F  ADDITIONAL DATASET ANALYSES

We have thoroughly analyzed the different aspects of data samples in our dataset, EQA-MX. We visualize the output distribution for all EQA tasks, as well as the object locations with respect to different spatial relations/locations, and the most frequent words found in our dataset.

### F.1  TASK OUTPUT DISTRIBUTIONS

As shown in Figs. 6,7 we balance outputs of our task distributions where possible in order to ensure the EQA-MX dataset is not biased. For the OG and POG tasks, the output distribution of all 52 categories is balanced to ensure models do not bias a particular object. Additionally, in Fig. 6, all binary tasks (EP, OAC, and PG) contain a $50/50$ split between *yes* and *no* answers. Because the CAESAR simulator randomly generates scenes populated with objects, the OC and OAC tasks do not have even task distributions. This can be explained by these tasks involving observed characteristics in scenes where some characteristics are more common than others. For example, since the max number of objects that can be generated in a scene is 10, the probability of an object have 9 objects to the left of it is much lower than the probability of an object having 2 objects to the left of it. Similarly, certain colors are more common in objects inside of the CAESAR simulator. These distributions are made more apparent in Fig. 8 (we report macro accuracy for models trained on these tasks).

### F.2  OBJECT LOCATIONS ANALYSES

We visualize object locations inside the EQA-MX dataset to show how different spatial relations have/don't have bias (Fig. 9). Particularly, since one of our contributions is the creation of the shelf environment, we show how since its visual views are aligned certain visual cues have bias.

