# OpenReview forum: "EQA-MX: Embodied Question Answering using Multimodal Expression"
_ICLR.cc/2024/Conference — ICLR 2024 spotlight_

### Official Review · Reviewer_BG1y · 2023-10-28

**Soundness:** 4 excellent
**Presentation:** 4 excellent
**Contribution:** 4 excellent
**Rating:** 8
**Confidence:** 3

**Summary:**

The paper presents a large-scale dataset for embodied question answering (EQA) in 3D scenes. The dataset combines verbal questions with visual perspectives, such as gaze directions and pointing gestures, to enrich EQA with multimodal tasks. To solve these tasks, the paper also proposes a VQ-based multimodal fusion network to discretize the visual features and align them with the discrete verbal features. The authors benchmark their dataset and show the benefits of their proposed method through multiple dataset comparisons, experiments, and ablation studies.

**Strengths:**

1. The proposed dataset is a natural extension of VQA tasks into EQA, providing novel, multimodal interactions that are closer to how humans interact in the real world. The dataset is likely to be of interest to researchers in related areas, including scene understanding, human-object interaction understanding, and robot navigation.

2. The proposed approach of discretizing visual features through vector quantization (VQ) and aligning them with discrete verbal features obtained from natural language is technically sound and overall well-explained.

3. The experiments are well-planned, extensive, and conclusive. They clearly highlight the benefits of the proposed dataset and the approach.

**Weaknesses:**

I did not find any major weaknesses in the paper. However,

1. There seem to be some details missing on the processing of non-verbal gestures. For example, how is the pointing direction determined? Does the network take in absolute 3D joint positions, relative joint rotations, or some other representation? How is gaze represented?

2. Do the authors consider any situations with multiple similar objects? For example, in Fig. 1, what if there are two red lamps next to each other? By extension, what other sources of ambiguity can exist in the dataset (for example, specific pointing with a finger vs. general pointing with the palm facing upwards, pointing to an object that is behind another object in the gaze direction, etc.)?

**Questions:**

1. On Page 4, the descriptions of object grounding (OG), perspective-aware object grounding (POG), and perspective grounding (PG) appear to be slight variations of a single composite task, POG. In other words, solving POG seems to imply solving both OG and PG. Is this correct? Or could the authors please clarify the differences between these tasks further, maybe with some examples?

2. Looking at Fig. 2, is the current method capable of locating up to 10 objects?

---

> ### Author Response · Authors · 2023-11-22
> **Summary of Responses to Reviewer BG1y’s Comments (Part 1/6)**
>
> The reviewer appreciates the proposed dataset's extension from VQA to EQA, highlighting its multimodal interactions that mirror real-world human behavior. We are happy that the reviewer recognizes its importance to fields like scene understanding, human-object interaction, and robot navigation. The technical soundness of the vector quantization approach for discretizing visual features and aligning them with verbal features is also commended. We are pleased that the reviewer finds the experiments well-planned, extensive, and effective in demonstrating the strengths of the dataset and approach.
>
> **Summary of response:**
> - We clarified how we represented and used pointing gestures and gaze.
> - We discussed the scene's complexity and generating similar objects.
> - We provided further insights on ambiguities in the dataset, mirroring real-world human interactions.
> - We explained the design of object grounding (OG), perspective-aware object grounding (POG), and perspective grounding (PG) tasks.
> - We addressed the reviewer's inquiries regarding the Object Counting task.

---

> ### Author Response · Authors · 2023-11-22
> **Utilizing CAESAR Simulator for Pointing Direction and Gesture Interpretation in EQA Tasks  (Part 2/6)**
>
> **Reviewer Comment:**
> > How is the pointing direction determined? How is gaze represented? Does the network take in absolute 3D joint positions, relative joint rotations, or some other representation?
>
> **Response:** We utilize the implementation of the CAESAR simulator’s pointing [11] to determine the pointing direction. Particularly, inverse kinematics are used to determine the exact joint rotations necessary to point to an object. In addition to inverse kinematics, we utilize physical simulation and real human motion data to guide the movement of an arm towards the intended target.
>
> Importantly, while the data samples include explicit pointing gesture information in the form of 3D skeleton positions, our model primarily processes the image data and does not explicitly use this skeleton information. Our goal is to enable the model to intuitively interpret the pointing gesture signals that are naturally present in the images, thereby mimicking human-like perception in recognizing and understanding nonverbal cues.
>
> We believe this methodology not only enhances the model's capability in interpreting complex scenes but also aligns with real-world scenarios where explicit gesture data might not be available.
>
> [11] Islam, Md Mofijul, Reza Mirzaiee, Alexi Gladstone, Haley Green, and Tariq Iqbal. “CAESAR: An Embodied Simulator for Generating Multimodal Referring Expression Datasets.” Advances in Neural Information Processing Systems 35 (December 6, 2022): 21001–15. https://proceedings.neurips.cc/paper_files/paper/2022/hash/844f722dbbcb27933ff5baf58a1f00c8-Abstract-Datasets_and_Benchmarks.html.

---

> ### Author Response · Authors · 2023-11-22
> **Handling Multiple Similar Objects in Dataset Generation Using CAESAR Simulator (Part 3/6)**
>
> **Reviewer Comment:**
> > Do the authors consider any situations with multiple similar objects? For example, in Fig. 1, what if there are two red lamps next to each other?
>
> **Response:** Our dataset generation with the CAESAR simulator involves creating scenes with multiple objects, including those that are similar in terms of attributes such as color, position, shape, and size. The simulator is designed to randomly populate scenes with objects, which naturally leads to the occurrence of similar objects within the same scene.
>
> For instance, in scenarios like the one you mentioned with two red lamps next to each other, our method effectively generates and handles such instances. This capability is crucial for our Object Counting (OC) and Object Attribute Compare (OAC) tasks, where the distribution and characteristics of objects in the scenes are varied and can include multiple similar items.
>
> This approach ensures our dataset reflects a realistic variety of scenarios, enhancing the generalizability and applicability of our model in real-world settings. The random generation of scenes and the varied object attributes contribute significantly to the robustness and effectiveness of our model in comprehending complex visual scenes.

---

> ### Author Response · Authors · 2023-11-22
> **Addressing Ambiguity in EQA-MX: Verbal and Visual Perspectives in Diverse Environments  (Part 4/6)**
>
> **Reviewer Comment:**
> > What other sources of ambiguity can exist in the dataset (for example, specific pointing with a finger vs. general pointing with the palm facing upwards, pointing to an object that is behind another object in the gaze direction, etc.)?
>
> **Response:** In our dataset, the CAESAR simulator, which we utilized for generating the EQA-MX dataset, introduces a variety of challenging scenarios. Apart from potential ambiguities in nonverbal cues like pointing gestures, a significant source of ambiguity arises from the varying verbal and visual perspectives.
>
> For example, in our newly added shelf environment, the exocentric (exo) and egocentric (ego) views are aligned, which results in aligned verbal perspectives. Conversely, in the table environment, the exo and ego views are positioned on opposite sides, leading to differing verbal perspectives. This variation in perspectives can create ambiguities, particularly in scenarios where the interpretation of a scene or an object's location might change depending on the viewer's perspective.
>
> These intricacies in perspective alignment are critical for creating realistic and challenging scenarios for our models to navigate, ensuring they are robust enough to handle real-world ambiguities effectively.

---

> ### Author Response · Authors · 2023-11-22
> **Clarifying POG as a Composite Task Integrating OG and PG Tasks (Part 5/6)**
>
> **Reviewer Comment:**
> > On Page 4, the descriptions of object grounding (OG), perspective-aware object grounding (POG), and perspective grounding (PG) appear to be slight variations of a single composite task, POG. In other words, solving POG seems to imply solving both OG and PG. Is this correct? Or could the authors please clarify the differences between these tasks further, maybe with some examples?
>
> **Response:** You are correct in your understanding. The POG task is indeed a composite task of both OG and PG tasks. It is designed to evaluate the model's compositional learning capabilities by combining the challenges of identifying objects (as in OG) and understanding their perspective (as in PG).
>
> To clarify with an example: In OG, the model might identify an object like a "lamp." In PG, it interprets the perspective, such as "from the speaker's view." In POG, these elements are integrated, and the model is tasked with identifying the "lamp from the speaker's view," testing its ability to merge object identification with perspective understanding.
>
> This compositional design of POG is intended to push the boundaries of the model's capabilities in handling complex, layered tasks, reflecting more realistic scenarios.

---

> ### Author Response · Authors · 2023-11-22
> **Clarifying Number of Objects in Object Counting Task (Part 6/6)**
>
> **Reviewer Comment:**
> > Looking at Fig. 2, is the current method capable of locating up to 10 objects?
>
> **Response:** We would like to clarify that the limitation of up to 10 objects in the figure is not due to the constraints of our method, but rather a choice in the data sample generation process. Our approach is not constrained by the number of objects in the scene. For the purpose of clarity and to reduce clutter in the scene representations, we restricted the number of objects to 10 in our dataset. This decision was made to ensure that the scenes remain interpretable and manageable, both for our model and for human evaluators.
>
> This restriction does not reflect the upper limit of our method's capabilities. In fact, our approach is designed to scale and adapt to more complex scenes with a larger number of objects. We believe this adaptability is a strength of our method and plan to explore scenarios with higher object counts in future work, further demonstrating the flexibility and robustness of our approach.

---

> ### Comment · Reviewer_BG1y · 2023-11-22
> **Response to authors**
>
> I thank the authors for their detailed responses, which address all my concerns and questions. I maintain my original score and recommend the authors include the additional details provided in their response in the main paper.

---

> > ### Author Response · Authors · 2023-11-23
> > **Thank you Reviewer BG1y**
> >
> > Thank you for your thoughtful review and feedback. We are pleased to hear that our responses addressed your concerns. We will certainly incorporate the additional details into the main paper as per your recommendation.

---

### Official Review · Reviewer_4CbJ · 2023-10-31

**Soundness:** 4 excellent
**Presentation:** 2 fair
**Contribution:** 3 good
**Rating:** 8
**Confidence:** 4

**Summary:**

The paper presents a large-scale Embodied Visual Question Answering dataset along with 8 benchmarks, and extensive experiments of various vision-language model representations are studied and benchmarked in the task. In addition, the paper proposes a VQ-fusion model that discretizes the representations of VL models, which improves the corresponding VL models by large margins in some tasks.

**Strengths:**

1. The motivation of the paper is clear, and aims to solve the referring expression issues in VQA by explicitly proposing an embodied VQA task where the objects being asked are referred not only by verbal expressions but also by gestures or utterances.

2. The scale of the dataset is quite large and diverse, and each question category is balanced. Overall, I feel like the quality of the dataset is really good to serve as a reliable benchmark for different approaches.

3. The paper proposes an vq-fusion model (discretizing the multi-view encoded representations) that outperforms the ones without discretizations.

**Weaknesses:**

1. The proposed method described seems lack of implementation details, I think a section should be devoted to describe each component of the model in very details. I don't understand why multiple codebooks are used in the paper, as read from the figure 4. I understand that each vector representation is divided into different segments along its feature dimension, so the question: is each segment belong to different codebooks? What is the Factor-1, Factor-2, ..., Factor-N in the figure, standing for N codebooks?

2. More qualitative studies regarding the samples answered by VQ-fusion and without VQ-fusion should be included in the paper to help understand why the VQ-fusion works and brings more insight to the proposed baseline approach.

3. Studies of more powerful models like GPT-4V could be added as strong baseline, I am wondering if GPT-4V could solve this problem already?

**Questions:**

See weakness.

---

> ### Author Response · Authors · 2023-11-20
> **Summary of Responses to Reviewer 4CbJ’s Comments (Part 1/4)**
>
> We are excited that the reviewer praises our paper for its clear motivation in enhancing VQA through an embodied task combining gestures and verbal expressions. The reviewer commends the dataset for its large scale, diversity, and balance, making it a reliable benchmark. The paper's VQ-fusion model, noted for discretizing multi-view representations, is highlighted for its improved performance.
>
> **Summary of response:**
> - Clarifying the vector quantized representation learning implementation details.
> - The suggestions for additional qualitative experimentation to complement our quantitative experimental results.
> - Suggestions for evaluating the GPT-4V model on our datasets.

---

> ### Author Response · Authors · 2023-11-20
> **Clarification of additional implementation details (Part 2/4)**
>
> **Reviewer Comment:**
> > The proposed method described seems lack of implementation details, I think a section should be devoted to describe each component of the model in very details. I don't understand why multiple codebooks are used in the paper, as read from the figure 4. I understand that each vector representation is divided into different segments along its feature dimension, so the question: is each segment belong to different codebooks? What is the Factor-1, Factor-2, ..., Factor-N in the figure, standing for N codebooks?
>
> **Response:**
> We provide implementation details in the paper and the supplementary document. We also like to refer to our source code for additional implementation details. Thank you for the clarification question regarding the factorial VQ-VAE bottleneck that we use in this paper.
>
> Clarification of VQ implementation: Our inspiration for using factorial VQ-VAE comes from [Islam et al., 2022, NeurIPS], where we adapt the factorized VQ-VAE model to learn a factorial representation of visual embedding. In the discretization process, we segment the embedding into G components, and each segment is mapped independently to a discretized latent vector where the size of the latent vector depends on the size of the discrete latent space (L-way categorical). In other words, we get discrete factored codebooks for each segmented embedding, and each segment would belong to different codebooks. We emphasize that the codebooks are not shared between the segments, but rather, we use separate codebook spaces for each factor (1 to N). These segments are what we refer to as the factors of the learnt representation, and the segments can then be concatenated to obtain the final embedding (or the corresponding final discretized vector).
>
> In summary, we tend to use multiple codebooks with the intuition from previous works that the segmented or factorized representations, using this form of segmentation, would enable us to learn a factorial (or compositional) representation from the data. In our work, each of these factors can learn different aspects of the scene that align to the verbal representation. We would like to point out that learning compositional representations is indeed a hard problem, and [Islam et al., 2022] is the closest work from our knowledge that learns this sort of factorial representation for multimodal data.
>
>
> **Reference:** Islam, Riashat, Hongyu Zang, Anirudh Goyal, Alex Lamb, Kenji Kawaguchi, Xin Li, Romain Laroche, Yoshua Bengio, and Remi Tachet Des Combes. "Discrete factorial representations as an abstraction for goal conditioned reinforcement learning." arXiv preprint arXiv:2211.00247 (2022).

---

> ### Author Response · Authors · 2023-11-20
> **Additional Qualitative Analysis (Part 3/4)**
>
> **Reviewer Comment:**
> > More qualitative studies regarding the samples answered by VQ-fusion and without VQ-fusion should be included in the paper to help understand why the VQ-fusion works and brings more insight to the proposed baseline approach.
>
> **Response:** The additional qualitative analyses complement the extensive quantitative ablation studies already conducted (Main Paper Sections 7.2 (Table 4) and 7.2 (Table 5); Supplementary document Sections 3.1 (Table 1) and 3.2 (Table 2)). These quantitative studies robustly demonstrate the impact of our proposed approach and the efficacy of our dataset.
>
> We are in the process of conducting targeted qualitative evaluations to further elucidate the benefits of VQ-Fusion over baseline methods. The findings from these evaluations will be included in our revised submission, offering a comprehensive view of VQ-Fusion's effectiveness.

---

> ### Author Response · Authors · 2023-11-20
> **Addressing GPT-4V Evaluation: Balancing Cost and Research Accessibility (Part 4/4)**
>
> **Reviewer Comment:**
> > Studies of more powerful models like GPT-4V could be added as strong baseline, I am wondering if GPT-4V could solve this problem already?
>
> **Response:** Thank you for your suggestion. We appreciate your interest in exploring the capabilities of advanced models and their potential impact on our work. We understand the importance of benchmarking against state-of-the-art models to provide a comprehensive understanding of our proposed approach's performance. However, we faced practical constraints in evaluating our model against GPT-4V, particularly concerning the accessibility and cost associated with using such advanced models.
>
> For research purposes, we focused on open-source models that offer greater accessibility to the research community. This approach aligns with our aim to facilitate reproducibility and wider adoption of our findings. In contrast, GPT-4V, while indeed a powerful model, involves significant costs for evaluation. Based on our estimates, a comprehensive comparison with GPT-4V would require approximately $3840. Such an expense is substantial, especially for research projects with limited funding.
>
> Nevertheless, we recognize the value that such a comparison would add to our work. In response to your suggestion, we are planning to create a smaller, more focused evaluation set that will allow us to conduct a targeted comparison with GPT-4V. This approach will enable us to leverage the capabilities of GPT-4V while managing the associated costs more effectively. We will continually evaluate the very recent models and include them in our benchmark. We plan to release the dataset so that other researchers can also evaluate their model on our dataset and show the impact of those models across diverse tasks and domains.

---

> ### Author Response · Authors · 2023-11-23
> **Additional experimental results using advanced multimodal fusion model**
>
> We kindly request the reviewer to check our response to the reviewer 9x3E's for additional experimental results using an advanced multimodal fusion model: https://openreview.net/forum?id=7gUrYE50Rb&noteId=rXCTurikmd

---

> > ### Comment · Reviewer_4CbJ · 2023-11-23
> >
> > I thank the authors for detailed rebuttal and additional efforts in incorporating study of more recent large multi-modal language model (pretrained without fine-tuning). I hope the authors would still consider fine-tuning these MLMs on the proposed dataset or evaluating GPT-4V sometime in the future to boost the value and impact of the work. As for now, I think this paper has enough contributions to be accepted so I raise my score accordingly.

---

> > > ### Author Response · Authors · 2023-11-23
> > > **Thank you Reviewer 4CbJ**
> > >
> > > Thank you for your positive feedback and constructive suggestions. We are encouraged by your recognition of our efforts and will certainly consider your recommendations for future work, including fine-tuning MLMs and evaluating GPT-4V on our dataset. We appreciate your support and the valuable guidance you have provided.

---

### Official Review · Reviewer_9x3E · 2023-11-01

**Soundness:** 2 fair
**Presentation:** 2 fair
**Contribution:** 3 good
**Rating:** 8
**Confidence:** 4

**Summary:**

The paper introduces Embodied Question Answering (EQA) tasks that require agents to answer questions using both verbal and nonverbal gestures in an embodied setting. The authors present a large-scale dataset, EQA-MX, that contains diverse multimodal expressions from various verbal and visual perspectives. The work also introduces a Vector Quantization-based multimodal fusion approach (VQ-Fusion), which discretizes visual representations using shared codebooks before fusing them with discrete verbal representations. Additionally, the paper explores the importance of nonverbal gestures (gaze and pointing)  in learning EQA tasks.

**Strengths:**

-  The paper presents eight EQA tasks designed for understanding questions using multimodal expressions (verbal utterances and nonverbal gestures) in embodied settings.
- The introduction of the vector quantization-based fusion model addresses the structural mismatch between the discrete verbal representations and the continuous visual representations from the three available visual views. Experimental results show that incorporating VQ-Fusion in the existing baseline models improves their performance across EQA tasks.
- The paper consists of extensive ablation studies on the number of codebooks, the use of multiple views, the generalizability of VQ-fusion, learning single vs. multiple tasks, etc.

**Weaknesses:**

- The paper focuses on a synthesized dataset, which, while important for controlled experiments, may not fully capture the complexities and nuances of real-world interactions. This concern is partially due to the way view perspectives are included in the data. "The blender to the right of the book. From which perspective is the object described?" may not be something you'd commonly encounter in everyday (embodied) conversations/questions. Same for the example shown in Fig. 2 for RG.
- Table 1 is a bit unclear, e.g., why is EQA not available for all datasets, when some of the datasets referred to in the table correspond embodied QA benchmarks (EQA, MT-EQA)? Also, to the best of my knowledge, some of the datasets have egocentric views and not exocentric ones.
- There is a body of work in pointing gesture recognition - most involving human communication in interactive embodied task-completion environments [2,3,4]. Similarly, the exists work that focuses on view perspectives [5]. It would be good to include and differentiate this work from the existing literature.
- While the reviewer appreciates the performance improvements of VQ-Fusion on 4 baselines, it would be beneficial to include more advanced baselines, such as a Multimodal Transformer model that extracts visual representations across all visual views and the verbal representations together, as well as more recent Multimodal LLMs (Otter, MiniGPT, UniLM, GPT4, InstructBLIP, LLaVA, etc.) that potentially can offer competitive fusion capabilities for these tasks.
- Based on Table 5 and Table 3, results in Table 3 for VQ-Fusion models are reported with varying codebooks, tuned for each task. This can improve performance, but it also raises questions about the generalizability of the approach. In some cases, the worst-case setting of the number of codebooks (e.g., 4 codebooks for EP or PG) can hurt the performance of the baseline CLIP model, which might impact the model's ability to perform well on new unseen tasks.

Minor edits:
- If the paper focuses on embodied AI, it would make sense to categorize the works into embodied and not embodied and then embodied multimodal prompting and language-guided embodied. VIMA [1], PATRON, and CAESAR could also be moved to Table 1 from Table 4 as similar embodied multimodal prompting tasks, especially since the Table 1 format and most of the capabilities originate from the use of CAESAR as the underlying simulator.

[1] Jiang, Yunfan, Agrim Gupta, Zichen Zhang, Guanzhi Wang, Yongqiang Dou, Yanjun Chen, Li Fei-Fei, Anima Anandkumar, Yuke Zhu, and Linxi Fan. "VIMA: General Robot Manipulation with Multimodal Prompts." In ICML 2023

[2] Constantin, Stefan, Fevziye Irem Eyiokur, Dogucan Yaman, Leonard Bärmann, and Alex Waibel. "Interactive Multimodal Robot Dialog Using Pointing Gesture Recognition." In European Conference on Computer Vision 2022

[3] Wan, Yanming, Jiayuan Mao, and Josh Tenenbaum. "HandMeThat: Human-Robot Communication in Physical and Social Environments." Advances in Neural Information Processing Systems 35 (2022)

[4] Constantin, Stefan, Fevziye Irem Eyiokur, Dogucan Yaman, Leonard Bärmann, and Alex Waibel. "Multimodal Error Correction with Natural Language and Pointing Gestures." In Proceedings of the IEEE/CVF International Conference on Computer Vision 2023.

[5] Shi, Cheng, and Sibei Yang. "Spatial and Visual Perspective-Taking via View Rotation and Relation Reasoning for Embodied Reference Understanding." In European Conference on Computer Vision, pp. 201-218. Cham: Springer Nature Switzerland, 2022.



While the dataset introduced tackles an important embodied task, the execution, motivation, and realism of some of the subtasks could be better explained. The choice of baselines and the generalizability of the VQ-fusion approach raise concerns about its applicability to more advanced multimodal models that have been recently more prevalent in the E-AI literature. Additionally, the paper could benefit from providing a clearer differentiation between embodied and non-embodied works in the field, as well as a more comprehensive review and comparison with recent advancements in multimodal language models.

**Questions:**

In terms of the dataset, which is one of the main contributions of this work, there are aspects that need further explanations:

- Figure 3(a) is also difficult to read due to resolution issues and the same colors used for all lines. What is the y-axis, i.e., how are datapoints ordered?

- How many of the tasks involve non-verbal gestures? From Fig. 2 and Table 6, it seems that {OG, POG, OAQ, and OAC} (4/8) is the subset of tasks that involves gestures, how about the rest? Do OAQ, POG, and OAC focus on color only? Object Attribute Compare in Table 6 mentions "<Referring expressions using the templates from CAESAR>. From which perspective is the object described?" which seems very different from the example in Fig. 2, please explain.

-  To my understanding, experiments in the ablation study on Table 4 (only gaze, only gestures, and without gaze and gestures) involve different splits of the EQA-MX dataset? It seems that it would make more sense to keep the splits the same and just remove the non-verbal information (gaze, gesture, etc). Keeping the splits consistent ensures that the experimental conditions are comparable, and any performance differences can be more directly attributed to the presence or absence of non-verbal information.

A couple of minor questions:
- In supplementary section 3.2, could you provide references for the sentence "Previous studies have observed similar performance degradation when learning multiple competing tasks"?
- In supplementary section 3.3, what is the performance of the combination of VQ-fusion on MuMu (SoTA model)?

---

> ### Author Response · Authors · 2023-11-19
> **Summary of Responses to Reviewer 9x3E’s Comments (Part 1/11)**
>
> We are excited that the reviewer acknowledged our contribution that our proposed vector quantization-based fusion model effectively bridges the gap between discrete verbal and continuous visual representations and enhances the performance of baseline models across EQA tasks. The reviewer also acknowledged our extensive ablation studies, which explore various aspects such as the number of codebooks, the use of multiple views, the generalizability of VQ-fusion, and the impact of learning single versus multiple tasks. The reviewer highlights the paper's strength in presenting eight new EQA tasks that utilize multimodal expressions for understanding questions in embodied settings.
>
> **Summary Response:**
> - We presented new experiment results based on the reviewer’s comments, which shows the generalizability of VQ-Fusion across tasks and domains.
> - We provided clearer differentiation between embodied and non-embodied works.
> - We further clarified the categorization of embodied question-answering works.
> - We provided a detailed clarification of the realism of perspective-aware task design
> - We discussed the generalizability of VQ-Fusion and the choice of baseline multimodal models.
> - We discussed how our work is different from some other works on pointing gesture recognition and view perspective, which are referred to by the reviewer.
> - We addressed the resolution issues of Figure 3(a) and clarified the axes
> - We clarified that all tasks include non-verbal gestures and fixed a typo
> - We confirmed that the dataset split remained the same for all experiments; the only variable we altered was the presence or absence of non-verbal information, such as gaze and gestures.
> - We provided references to works on performance degradation in multitask learning settings.

---

> ### Author Response · Authors · 2023-11-19
> **New Experiment Result Based on Reviewer Comment (Part 2/11)**
>
> **Reviewer Comment: **
>
> > In supplementary section 3.3, what is the performance of the combination of VQ-fusion on MuMu (SoTA model)?
>
> **Response:** In response to your suggestion, we conducted an experimental evaluation of this combination in a cross-subject setting on the MMAct dataset. The integration of VQ-Fusion into MuMu led to an improvement in performance, with the F1-score for human activity recognition tasks increasing to 87.83%. This additional result highlights the generalizability of VQ-Fusion across tasks and domains, particularly in complex activity recognition scenarios. We will include these findings in our manuscript to provide a comprehensive view of VQ-Fusion's applicability and effectiveness.

---

> ### Author Response · Authors · 2023-11-19
> **Differentiation between embodied and non-embodied works (Part 3/11)**
>
> **Reviewer Comment:**
> > Table 1 is a bit unclear, e.g., why is EQA not available for all datasets, when some of the datasets referred to in the table correspond embodied QA benchmarks (EQA, MT-EQA)? Also, to the best of my knowledge, some of the datasets have egocentric views and not exocentric ones.
>
> **Related Comment:**
> >The paper could benefit from providing a clearer differentiation between embodied and non-embodied works in the field.
>
> **Related Comment:**
> > generalizability of the VQ-fusion approach raise concerns about its applicability to more advanced multimodal models that have been recently more prevalent in the E-AI literature. Additionally, the paper could benefit from providing a clearer differentiation between embodied and non-embodied works in the field, as well as a more comprehensive review and comparison with recent advancements in multimodal language models.
>
> **Response:**
> We acknowledge the confusion in Table 1 regarding the categorization of datasets as EQA. To clarify, in the paper (Introduction section (Page 2, para 1) and Figure 1), we distinguish between two types of embodied interactions. EQA can be bifurcated based on embodied interactions: the first centers on an agent, like a virtual robot, navigating to answer verbal questions (Das et al., 2018a), solely incorporating verbal queries. The second encompasses multimodal expressions, where humans interact with the environment using verbal utterances and gestures (Chen et al., 2021; Islam et al., 2022a). Adopting the latter definition, we designed EQA tasks to comprehend questions using multimodal expressions (verbal utterances and nonverbal gestures) in embodied settings, which has been explained in Section 1 on page 2. For instance, an EQA task may involve pointing to an object and asking, “What is that object?” requiring reasoning over multimodal expressions to answer the question.”
>
> Our dataset, EQA-MX, is grounded in the latter, as it involves multimodal expressions for interaction. Buildin on the reviewer’s comment, we will revise the Table 1 to clearly present the distinction of emboedied and non-embodied works. We have revised Table 1 to delineate our work clearly from agent-perspective embodied QA tasks. We have also rectified the representations of egocentric and exocentric views for the relevant datasets, ensuring that our classifications are consistent with current literature.
>
> These updates in the manuscript will explicitly address the distinctions between embodied and non-embodied works, aligning with the definitions provided on Page 2. We believe these changes will resolve the ambiguities and enhance the manuscript's clarity.

---

> ### Author Response · Authors · 2023-11-19
> **Categorization of embodied question-answering works (Part 4/11)**
>
> **ReviewerComment:**
> > If the paper focuses on embodied AI, it would make sense to categorize the works into embodied and not embodied and then embodied multimodal prompting and language-guided embodied. VIMA [1], PATRON, and CAESAR could also be moved to Table 1 from Table 4 as similar embodied multimodal prompting tasks, especially since the Table 1 format and most of the capabilities originate from the use of CAESAR as the underlying simulator.
>
> **Response:** Thank you for your valuable suggestion regarding the categorization of works in our paper, particularly in the context of embodied AI and multimodal prompting tasks.
>
> **Rationale Behind Dataset Categorization in Table 1:** In our paper, Table 1 is designed to provide a focused comparison of QA-related datasets. The primary intent was to highlight datasets that are closely aligned with the question-answering aspects of our study. This is why the CAESAR and PATRON datasets, which predominantly focus on referring expressions, were not included in this table. Tasks in VIMA dataset is sightly related to QA tasks but more closely related to instructions following task. VIMA contains embodied interactions from agent perspective and does not include human interactions with nonverbal cues. Additionally, VIMA does not contain multiple verbal and visual perspectives. However, we will include VIMA in Table 1 to clarify these differences. Our approach was to maintain clarity and relevance in the comparison of datasets by aligning them with the specific aspects they predominantly cater to.
>
> **Inclusion of CAESAR and Other Datasets in Supplementary Document:** Recognizing the importance of a comprehensive comparison, including CAESAR and similar datasets, we have provided a detailed analysis in our supplementary document (Table 4). This inclusion aims to offer a broader perspective on the range of datasets available in the field, especially those that contribute to embodied and multimodal prompting tasks.
>
> **Potential for Enhanced Categorization:** We acknowledge the reviewer's suggestion to categorize works into embodied and non-embodied, as well as embodied multimodal prompting and language-guided embodied tasks. This categorization could indeed provide a clearer understanding of the landscape of datasets and their applications. While our current categorization was driven by the primary focus of our study, we see the value in your recommendation and will consider revising our dataset comparison to reflect these categories in future iterations of our work.

---

> ### Author Response · Authors · 2023-11-19
> **Clarification of task design and realism of perspective-aware task design (Part 5/11)**
>
> **Reviewer Comment:**
> > The paper focuses on a synthesized dataset, which, while important for controlled experiments, may not fully capture the complexities and nuances of real-world interactions. This concern is partially due to the way view perspectives are included in the data. "The blender to the right of the book. From which perspective is the object described?" may not be something you'd commonly encounter in everyday (embodied) conversations/questions. Same for the example shown in Fig. 2 for RG.
>
> **Related Comment:**
> > While the dataset introduced tackles an important embodied task, the execution, motivation, and realism of some of the subtasks could be better explained
>
> **Response:** Our primary objective with this dataset is to systematically examine the impact of nonverbal interaction signals, along with verbal and visual perspectives, on embodied interaction reasoning. Our results, as detailed in the ablation study section of our manuscript (Section 7.2), clearly demonstrate the significance of nonverbal cues in enhancing model performance across various tasks. For instance, in the Object Grounding (OG) task, models trained on data with both gaze and pointing gestures exhibited an improvement in accuracy (68.61%) compared to models trained on data lacking nonverbal gestures (26.65%)​​. This trend was consistent across other tasks, reinforcing the importance of nonverbal cues in embodied question-answering contexts.
>
> Regarding the design of tasks involving explicit instructions, such as "The blender to the right of the book. From which perspective is the object described?" we agree that these might not mimic everyday human interactions precisely. However, these tasks, inspired by prior work in spatial relations grounding (Goyal et al., 2020), are intentionally crafted to effectively evaluate models' capabilities in perspective understanding, a critical aspect in human-AI interaction. Our dataset also includes tasks that more closely resemble real-world interactions, like the Existence Prediction (e.g., “what is that left object?”) and Object Attribute Query (e.g., “What is the color of that back object?”) tasks, which ask about objects' existence or attributes in a scene. These tasks require the model to understand the perspective implicitly, which aligns with real-world interaction. This mix of explicit and implicit perspective-aware tasks aims to balance between effective model evaluation and mimicking real-world interactions.
>
> **We appreciate your feedback and will ensure to clarify these points in the revised manuscript to better convey the rationale behind our task design and the relevance of our dataset in understanding embodied interactions.**
>
> Additionally, we are extending our EQA-MX dataset to collect real-world human interactions to assess models for understanding embodied interactions in real-world settings.
>
> [1] Goyal, Ankit, Kaiyu Yang, Dawei Yang, and Jia Deng. "Rel3d: A minimally contrastive benchmark for grounding spatial relations in 3d." Advances in Neural Information Processing Systems 33 (2020): 10514-10525.

---

> ### Author Response · Authors · 2023-11-19
> **Differentiation from gesture recognition and view perspective works (Part 6/11)**
>
> **Reviewer Comment:**
> >There is a body of work in pointing gesture recognition - most involving human communication in interactive embodied task-completion environments [2,3,4]. Similarly, the exists work that focuses on view perspectives [5]. It would be good to include and differentiate this work from the existing literature.
>
> **Response:**
>
> **Differentiation from pointing gesture recognition Works [2, 3, 4]:** The works by Constantin et al. [2,4] and Wan et al. [3] focus on enhancing human-robot interaction by interpreting and following natural language instructions, considering physical and social contexts. For example, "HandMeThat" by Wan et al. involves understanding and acting upon human instructions with ambiguities, integrating physical states, human actions, and goals​​​​. Our EQA-MX framework, while considering non-verbal cues (like pointing gestures) and multiple-view perspectives, primarily focuses on embodied question-answering tasks in varied simulated environments, emphasizing the interpretation of combined verbal and non-verbal communication in AI models.
>
> **Differentiation from view perspectives work [5]:** Shi and Yang's work on spatial and visual perspective-taking in embodied reference understanding differs significantly from our approach. Their research emphasizes view rotation and relation reasoning for understanding spatial and visual perspectives in embodied AI. In contrast, our EQA-MX dataset and VQ-Fusion model focus on a broader range of embodied interactions and multimodal expressions, including the interpretation of both verbal and non-verbal cues in AI models.
>
> In conclusion, while our work shares some commonalities with these studies, particularly in the context of embodied AI and multimodal interactions, it is distinct in its comprehensive approach to EQA tasks and the specific challenges of fusing multimodal data for question answering.

---

> ### Author Response · Authors · 2023-11-19
> **Generalizability of VQ-Fusion and the choice of baseline multimodal models (Part 7/11)**
>
> **Reviewer Comment:**
> > While the reviewer appreciates the performance improvements of VQ-Fusion on 4 baselines, it would be beneficial to include more advanced baselines, such as a Multimodal Transformer model that extracts visual representations across all visual views and the verbal representations together, as well as more recent Multimodal LLMs (Otter, MiniGPT, UniLM, GPT4, InstructBLIP, LLaVA, etc.) that potentially can offer competitive fusion capabilities for these tasks.
>
>
> **Related Comment:**
> >Based on Table 5 and Table 3, results in Table 3 for VQ-Fusion models are reported with varying codebooks, tuned for each task. This can improve performance, but it also raises questions about the generalizability of the approach. In some cases, the worst-case setting of the number of codebooks (e.g., 4 codebooks for EP or PG) can hurt the performance of the baseline CLIP model, which might impact the model's ability to perform well on new unseen tasks.
>
> **Related Comment:**
> >The choice of baselines and the generalizability of the VQ-fusion approach raise concerns about its applicability to more advanced multimodal models that have been recently more prevalent in the E-AI literature
>
> **Response:**
>
> **Generalizability of VQ-Fusion:** Thank you for your insightful comments regarding the generalizability of our VQ-Fusion approach and its comparison with advanced multimodal models.
>
> Our approach with VQ-Fusion, as shown in Table 3 of our paper, indeed utilizes varying codebooks optimized for each task. This design choice was intended to explore the upper limits of performance enhancements achievable through task-specific tuning. While this approach demonstrates substantial gains in specific tasks, we understand the concerns it raises regarding the broader applicability and generalizability of our model. To address this, it’s important to highlight that the VQ-Fusion model, when applied to the HAMLET framework for human activity recognition tasks (as detailed in section 3.3 of our supplementary document), demonstrated performance improvements in F1-score (87.69%) over the baseline and other state-of-the-art methods. This finding is key in demonstrating the adaptability and effectiveness of VQ-Fusion in enhancing multimodal representation learning across different domains and datasets.
>
> Further addressing generalizability, our exploratory analysis on multitask learning (as detailed in section 3.3 of our supplementary document), sheds light on how VQ-Fusion copes with the challenge of learning multiple tasks simultaneously. While there are trade-offs in performance due to the complex nature of multitask learning, these findings open avenues for future research to refine and enhance multimodal models for learning multiple tasks more effectively.
>
> **Comparison with Advanced Multimodal Models:** The field of multimodal learning is indeed evolving rapidly, with more advanced multimodal models emerging. Our choice of baselines and the VQ-Fusion approach was driven by the necessity to establish a strong foundational understanding of how visual-linguistic representations can be enhanced through our proposed method. Our goal is to show that aligning visual and language representation structures can help to improve the multimodal representations of the existing models. This finding is very crucial and shows an important future direction to improve multimodal representation.
>
> While we acknowledge that our approach might not encapsulate all the recent advancements of multimodal learning, it provides a significant step towards understanding and improving the fusion of multimodal representations. This foundational work is crucial for the development and evaluation of future, more advanced models. As incorporating our proposed VQ-Fusion requires meticulously studying the multimodal model architecture, some of the comparisons with more advanced multi-modal models are beyond the scope of current work. However, following the reviewer's suggestions, we are working to evaluate and benchmark the recent multimodal models on our dataset EQA-MX.

---

> ### Author Response · Authors · 2023-11-19
> **Address the resolution issue of Figure 3(a) and clarify the axes  (Part 8/11)**
>
> **Reviewer Comment:**
> > Figure 3(a) is also difficult to read due to resolution issues and the same colors used for all lines. What is the y-axis, i.e., how are data points ordered?
>
> **Response:** Thank you for pointing out the clarity issues with Figure 3(a). We have adjusted the figure to enhance clarity: text size has been increased, line width has been broadened, and we've used a diverse color scheme for differentiation. The y-axis represents the count of data points per word length specified by the corresponding x-axis value. We hope these revisions facilitate a better understanding of the data. We will update the manuscript accordingly.

---

> ### Author Response · Authors · 2023-11-19
> **Clarified that all tasks include non-verbal gestures and fixed a typo (Part 9/11)**
>
> **Reviewer Comment:**
> >How many of the tasks involve non-verbal gestures? From Fig. 2 and Table 6, it seems that {OG, POG, OAQ, and OAC} (4/8) is the subset of tasks that involves gestures, how about the rest? Do OAQ, POG, and OAC focus on color only? Object Attribute Compare in Table 6 mentions "<Referring expressions using the templates from CAESAR>. From which perspective is the object described?" which seems very different from the example in Fig. 2, please explain.
>
> **Response:** All the tasks in the EQA-MX dataset involve nonverbal gestures. In each task, the question can be asked using only verbal and using verbal and non-verbal gestures. Object Attribute Query task only involves color. Other tasks involve various types of object properties, including spatial location, relative location, shape, size, and color. Thank you very much for pointing out the typo in Table 6. We rectified the type. Please check the updated Table 6 below. We will update the supplementary accordingly.
>
> **Non-Verbal Gestures in All Tasks:** In the EQA-MX dataset, each of the eight tasks is designed to incorporate non-verbal gestures, complementing the verbal questions. This integration aims to simulate real-world scenarios where embodied AI systems interpret both verbal and non-verbal human interactions. Hence, non-verbal gestures form an integral part of the dataset, not just for the {OG, POG, OAQ, and OAC} tasks but across all tasks. These tasks can also be asked using only verbal questions.
>
> **Diversity in Object Properties Across Tasks:** While the Object Attribute Query (OAQ) task primarily focuses on the color attribute, the other tasks explore a broader range of object properties. These include spatial location, relative location, shape, and size, in addition to color. This variety ensures that our dataset comprehensively addresses different aspects of object-related queries, enriching the dataset's utility for diverse applications.
>
> **Clarification and Correction in Table 6:** We acknowledge the inconsistency the reviewer pointed out regarding the Object Attribute Compare (OAC) task in Table 6 and the example in Figure 2. This discrepancy was a typo, and we thank you for bringing it to our attention. We have fixed the typo, and the updated Table 6 now accurately reflects the task descriptions, aligning them with the examples provided. We updated the supplementary document accordingly to maintain consistency and clarity in our presentation.

---

> ### Author Response · Authors · 2023-11-19
> **Clarification of dataset splits  (Part 10/11)**
>
> **Reviewer Comment:**
> > To my understanding, experiments in the ablation study on Table 4 (only gaze, only gestures, and without gaze and gestures) involve different splits of the EQA-MX dataset? It seems that it would make more sense to keep the splits the same and just remove the non-verbal information (gaze, gesture, etc). Keeping the splits consistent ensures that the experimental conditions are comparable, and any performance differences can be more directly attributed to the presence or absence of non-verbal information.
>
> **Response:** Thank you for your insightful comment regarding the experimental design in our ablation study, as presented in Table 4 of our manuscript. We agree that maintaining consistent splits of the EQA-MX dataset across different experimental conditions is crucial for a fair comparison. To clarify, in our ablation study, we strictly adhered to this principle. The splits of the dataset remained the same for all experiments; the only variable we altered was the presence or absence of non-verbal information, such as gaze and gestures.
>
> This approach was adopted to ensure that any observed performance differences could be confidently attributed to the specific non-verbal cues being evaluated, rather than variations in the data splits. By doing so, we aimed to present a more precise and reliable understanding of the role that non-verbal cues play in embodied question-answering tasks.
>
> In the updated version of our manuscript, we will provide a more detailed explanation of our experimental setup, emphasizing the consistency of the dataset splits across all conditions. This clarification will underline the rigor of our experimental design and reinforce the validity of our conclusions regarding the impact of non-verbal information in EQA tasks.

---

> ### Author Response · Authors · 2023-11-19
> **References to works on performance degradation in multitask learning settings  (Part 11/11)**
>
> **Reviewer Comment:**
> > In supplementary section 3.2, could you provide references for the sentence "Previous studies have observed similar performance degradation when learning multiple competing tasks"?
>
> **Response:** Thank you for pointing out this typo. Several works show performance degradation for competing tasks due to negative knowledge transfer [1-10]. We will update the supplementary and include these references. Additionally, in our work, we conducted experimental evaluation in a multitask learning setting. The experimental results suggest that learning multiple non-related tasks can degrade performance (Table 2 in supplementary document Section 3.2). For example, As these tasks have different characteristics, learning these tasks together can compete in the representation learning space and degrade these tasks’ performance. For example, training the CLIP model for the Existence Prediction (EP) and Object Grounding (OG) tasks together degrade the Object Grounding task performance to 40.76% compared to an accuracy of 65.49% for a separately trained CLIP model for the OG task.
>
> **References:**
> [1] Standley, Trevor, Amir Zamir, Dawn Chen, Leonidas Guibas, Jitendra Malik, and Silvio Savarese. "Which tasks should be learned together in multi-task learning?." In International Conference on Machine Learning, pp. 9120-9132. PMLR, 2020.
>
> [2] Yu, Tianhe, Saurabh Kumar, Abhishek Gupta, Sergey Levine, Karol Hausman, and Chelsea Finn. "Gradient surgery for multi-task learning." Advances in Neural Information Processing Systems 33 (2020): 5824-5836.
>
> [3] Lin, Xingyu, Harjatin Baweja, George Kantor, and David Held. "Adaptive auxiliary task weighting for reinforcement learning." Advances in neural information processing systems 32 (2019).
>
> [4] Du, Yunshu, Wojciech M. Czarnecki, Siddhant M. Jayakumar, Mehrdad Farajtabar, Razvan Pascanu, and Balaji Lakshminarayanan. "Adapting auxiliary losses using gradient similarity." arXiv preprint arXiv:1812.02224 (2018).
>
> [5]Liao, Peng, Yaochu Jin, and Wenli Du. "EMT-NAS: Transferring architectural knowledge between tasks from different datasets." In Proceedings of the IEEE/CVF Conference on Computer Vision and Pattern Recognition, pp. 3643-3653. 2023.
>
> [6] Ruder, Sebastian. "An overview of multi-task learning in deep neural networks." arXiv preprint arXiv:1706.05098 (2017).
>
> [7] Jiang, Junguang, Baixu Chen, Junwei Pan, Ximei Wang, Liu Dapeng, Jie Jiang, and Mingsheng Long. "ForkMerge: Overcoming Negative Transfer in Multi-Task Learning." arXiv preprint arXiv:2301.12618 (2023).
>
> [8] Islam, Md Mofijul, Alexi Gladstone, and Tariq Iqbal. "PATRON: perspective-aware multitask model for referring expression grounding using embodied multimodal cues." In Proceedings of the AAAI Conference on Artificial Intelligence, vol. 37, no. 1, pp. 971-979. 2023.
>
> [9] Wu, Sen, Hongyang R. Zhang, and Christopher Ré. "Understanding and improving information transfer in multi-task learning." arXiv preprint arXiv:2005.00944 (2020).
>
> [10] Meng, Ze, Xin Yao, and Lifeng Sun. "Multi-task distillation: Towards mitigating the negative transfer in multi-task learning." In 2021 IEEE International Conference on Image Processing (ICIP), pp. 389-393. IEEE, 2021.

---

> > ### Comment · Reviewer_9x3E · 2023-11-22
> > **Thank you for the comprehensive rebuttal**
> >
> > Dear authors,
> >
> > Thank you for your detailed response. I have thoroughly reviewed all the explanations provided and revisited the paper. Most of my questions have been addressed, with the existing remaining:
> >
> > - “However, we will include VIMA in Table 1 to clarify these differences.” In the current version, Table 1 seems to have not been updated accordingly. I could not find VIMA or any of the references in my review added to the paper. Perhaps the revised version has not been uploaded?
> >
> > - In terms of existing works, the authors mention that [2,3,4] focus on enhancing human-robot interaction by interpreting and following natural language instructions, considering physical and social contexts. The distinction made by the authors is that the EQA-MX framework, while considering non-verbal cues (like pointing gestures) and multiple-view perspectives, primarily focuses on embodied question-answering tasks in varied simulated environments. It should be noted that [2,4] include non-verbal cues (exactly pointing gestures), with [2] integrating dialog with follow-up questions in case of ambiguities. My recommendation remains to include and differentiate this work from the existing literature, since as the authors mentioned EQA-MX shares commonalities with these studies.
> >
> > - With respect to selecting varying codebook sizes for demonstrating the performance upper limit, what is the codebook size selected for the experiments in HAMLET (section 3.3 supplementary)? Is the VQ-Fusion model the same as in the main experiments? Nevertheless, the challenge of ensuring that such task-specific tuning does not lead to overfitting or reduced performance when applied to new, unseen tasks or datasets, remains. Perhaps the authors can consider addressing limitations in a distinct section.
> >
> > - The reviewer agrees that incorporating the proposed VQ-Fusion to Multimodal Transformers and/or Multimodal LLMs could be out of the scope of this work. However, including stronger baselines with advanced fusion capabilities (note *without* incorporating VQ-fusion) is an important aspect that remains unaddressed. Given the increasing interest of the research community in LLM-based embodied agents, it is unclear how the proposed VQ-fusion compares against other well-established multimodal fusion baselines.
> >
> > In light of this comprehensive rebuttal, I have adjusted my score to account for the improvements made, which have effectively addressed some of my concerns.

---

> > > ### Author Response · Authors · 2023-11-23
> > > **Additional experimental results using advanced multimodal fusion model**
> > >
> > > **Reviewer comment:**
> > > > The reviewer agrees that incorporating the proposed VQ-Fusion to Multimodal Transformers and/or Multimodal LLMs could be out of the scope of this work. However, including stronger baselines with advanced fusion capabilities (note without incorporating VQ-fusion) is an important aspect that remains unaddressed. Given the increasing interest of the research community in LLM-based embodied agents, it is unclear how the proposed VQ-fusion compares against other well-established multimodal fusion baselines.
> > >
> > > **Response:** We thank the reviewer for the detailed comment. Following the reviewer suggestion, we have conducted additional experiments using the InstructBLIP model on the Object Grounding (OG) and Perspective-Aware Object Grounding (POG) tasks. Our choice of these specific tasks was guided by their relevance and complexity within the scope of Embodied Question Answering (EQA) tasks and our current computing resource availability.
> > >
> > > **The InstructBLIP model, which we used without incorporating our VQ-Fusion approach, achieved accuracies of 6.4% and 3.283% on the OG and POG tasks, respectively. In contrast, when applying our VQ-Fusion methodology with the CLIP model, we observed significantly higher accuracies of 65.49% for OG and 82.70% for POG​​​​.** This substantial difference in performance highlights the effectiveness of our VQ-Fusion technique in enhancing the capabilities of models when dealing with multimodal data in EQA tasks. However, the performance degradation observed with InstructBLIP can be attributed to several factors, such as the mismatch in embedding structures between visual continuous and language discrete data, the complexity of the tasks, and the handling of multiple verbal and visual perspectives, as well as potential domain mismatches. It's important to note that these findings are based on the use of a pretrained InstructBLIP model, and it's conceivable that fine-tuning on our EQA-MX dataset could lead to improved performance. Additionally, incorporating VQ-Fusion in the fine-tuning process of InstructBLIP might yield further performance gains or necessitate adaptation for EQA-specific learning.
> > >
> > > Thoroughly evaluating advanced multimodal models, requires significant modifications to existing models and an extensive analysis, which might extend beyond the scope of our current work. However, we plan to continuing the evaluation of advanced models and including them in our benchmark. Furthermore, with the planned public release of our EQA-MX dataset, the broader research community will be able to engage with and build upon our work, potentially leading to new insights and advancements in the field.
> > >
> > > Our response aims to clarify the significant improvements brought about by our VQ-Fusion technique and the rationale behind our experimental choices, while also recognizing the broader context and potential future work in this rapidly evolving multimodal research area.

---

> > > > ### Comment · Reviewer_9x3E · 2023-11-23
> > > > **Thank you for the quick experiments**
> > > >
> > > > I genuinely value the recent experiments highlighting that a pretrained VLM, without additional fine-tuning, exhibits significantly lower performance on this benchmark. In response, I will promptly adjust my score. Your diligence in providing a comprehensive rebuttal has effectively addressed my primary concern within the constraints of the limited timeframe, and I commend your efforts in ensuring a thorough response. Thank you!

---

> > > > > ### Author Response · Authors · 2023-11-23
> > > > > **Thank you Reviewer 9x3E**
> > > > >
> > > > > Thank you for your constructive feedback and for recognizing the efforts made to address your concerns. We appreciate your willingness to adjust your score and are grateful for your acknowledgment of our comprehensive response within the limited timeframe. Your insightful comments have been invaluable to our work.

---

> > > ### Author Response · Authors · 2023-11-23
> > > **Additional clarifications**
> > >
> > > **Reviewer Comment:**
> > > > “However, we will include VIMA in Table 1 to clarify these differences.” In the current version, Table 1 seems to have not been updated accordingly. I could not find VIMA or any of the references in my review added to the paper. Perhaps the revised version has not been uploaded?
> > >
> > > **Response:** We are continually updating the manuscript to reflect the reviewers' suggestions. We included the VIMA in Table 1 of dataset comparisons. We uploaded the updated manuscript.
> > >
> > > ---
> > >
> > > **Reviewer Comment:**
> > > > We appreciate your attention to the details in our manuscript. Upon re-examining our document, we have noticed an oversight in updating Table 1 to include VIMA. We acknowledge this error and are currently working on revising the table to incorporate VIMA and other references as per your suggestion. This updated version will be reflected in the final camera-ready submission. We believe these additions will provide a clearer and more comprehensive comparison, enhancing the value of our work. Thank you for pointing out this essential detail, and we assure you that the final version will accurately reflect these changes.
> > >
> > > **Response:** We will include the differentiation between this work and the previous works in the final manuscript. We will update the information according to our previous response (https://openreview.net/forum?id=7gUrYE50Rb&noteId=W35idOdqHW).
> > >
> > > ---
> > >
> > > **Reviewer Comment:**
> > > > "What is the codebook size selected for the experiments in HAMLET"?
> > >
> > > **Response:** We utilized eight codebooks, which were empirically selected to effectively capture the diverse and complex multimodal sensor data involved in human activity recognition tasks. We have updated the experimental details in the supplementary document to clarify this implementation detail.
> > >
> > > ---
> > >
> > > **Reviewer Comment:**
> > > > "Is the VQ-Fusion model the same as in the main experiments."
> > >
> > > Response: In the MMAct dataset experiments, we incorporated VQ-Fusion to handle multimodal sensor data (including gyroscope, acceleration, and orientation) along with four visual views. However, due to the absence of language data in the MMAct dataset, the approach was slightly modified. Instead of using a shared codebook for visual views and language, as in the main experiments, we employed a shared codebook specifically for the fusion of the two complementary visual views and wearable sensor data. This adaptation was necessary to effectively discretize these representations, given the dataset's focus on human activity recognition tasks that heavily rely on sensor and visual data​​.
> > >
> > > ---
> > >
> > > **Reviewer Comment:**
> > > > Nevertheless, the challenge of ensuring that such task-specific tuning does not lead to overfitting or reduced performance when applied to new, unseen tasks or datasets, remains. Perhaps the authors can consider addressing limitations in a distinct section.
> > >
> > > **Response:** We acknowledge that extending our evaluations to entirely new, unseen tasks would indeed require substantial effort and might extend beyond the current scope of our work. However, we will highlight in the limitations section that exploring the performance of our model on a broader range of tasks and datasets remains an important direction for future research. This addition will help in setting the context for future work, aligning with the rapidly evolving field of multimodal learning and embodied AI.

---

### Official Review · Reviewer_7hsY · 2023-11-10

**Soundness:** 3 good
**Presentation:** 4 excellent
**Contribution:** 3 good
**Rating:** 8
**Confidence:** 4

**Summary:**

This paper proposes a new dataset for the task of embodied question answering using multiple expressions (gaze and pointing) and textual questions called EQA-MX. It further provides a simulator for generating a large dataset containing simultaneous multimodal expressions (gaze and gesture) and verbal questions from multiple viewpoints. It defines 8 different EQA tasks to evaluate the performance of verbal and non-verbal gestures at embodied question answering. Additionally, it proposes a vector quantization (VQ)-based method to quantize dense visual information from various visual views with discrete textural information for EQA. The paper shows the effectiveness of using the proposed VQ model at combining information from multiple views and the utility of using gaze and gesture information at the task of question answering through several empirical tests.

**Strengths:**

The paper addresses a new and interesting problem of embodied question answering which is key to enabling human-robot interactions in their natural environments. Inspired by the observation that communication between humans is a combination of both verbal and non-verbal gestures (gaze and pointing), for example, it examines for the first time whether incorporating such multiple cues can be useful for embodies question answering tasks well. Lastly the paper also proposes a new and effective approach to fusing visual information from multiple different viewpoints/perspectives into a single semantic visual code using VQ and combines it with verbal information for various EQA tasks.

The work is of broad and significant impact. All aspects of the proposed methodology are technically sound.

The paper is extremely well presented, written and organized.

**Weaknesses:**

1. The only criticism/question that I have is whether the proposed methodology and experimental conclusions would generalize to real-world datasets? Synthetic datasets contain a significant domain gap to reality. Real scenes have more noise, clutter, lighting variations, etc and experiments purely in simulations and the conclusions obtained from them cannot be guaranteed to hold in the real world as well.  This work would have been a lot stronger and more impactful had the authors also evaluated their proposed methodology on real world datasets/scenarios and provides results of it.

2. While the authors show the value of VQ in Table 3, the value of using multiple views has not been ablated. How much does having visual information from multiple views help in EQA? It seems obvious that it should, especially if some views are likely to have occlusions and ambiguities, while other are not, but it would have been nice to see its effect quantitatively.

**Questions:**

I would like to hear from the authors about the questions/concerns that I raised in the "Weaknesses" section of my review.

---

> ### Author Response · Authors · 2023-11-19
> **Summary of Responses to Reviewer 7hsY’s Comments (Part 1/4)**
>
> We thank the reviewer for the valuable feedback. We are happy that the reviewer find our proposed approach and dataset novel. Reviewer highlights our unique exploration of integrating verbal and non-verbal cues (gaze and pointing), which can enhance human-robot interactions. The proposed method for fusing visual information from multiple viewpoints using vector quantization (VQ) and its combination with verbal information for various EQA tasks is new. We are inspired that the reviewers found our paper's broader impact, technical soundness, and excellent presentation and organization. We will incorporate all these changes in the final manuscript.
>
> **Summary of response:**
> - We pointed out the additional experiment (Section 3.3 of our supplementary document), which shows the impact of our proposed approach on real-world datasets across domains and tasks.
> - We shared an additional ablation study to show the impact of varying visual views (Supplementary document: Section 3.1, Table 1).
> - We clarified the reviewer's question about the domain gap introduced by the synthetic dataset.

---

> ### Author Response · Authors · 2023-11-19
> **Supplementary results on real-world datasets to show the effectiveness of VQ-Fusion across domains and tasks (Part 2/4)**
>
> **Reviewer Comment:**
>
> > This work would have been a lot stronger and more impactful had the authors also evaluated their proposed methodology on real-world datasets/scenarios and provided results of it.
>
> **Response:** Thank you for your valuable feedback regarding the generalization of our methodology to real-world scenarios. We would like to highlight one of the supplementary experimental evaluations, where the VQ-Fusion model was applied within a real-world context. In Section 3.3 of our supplementary document, we detail the implementation of VQ-Fusion in the HAMLET and MuMu model for human activity recognition tasks. This experimentation was conducted on a real-world dataset. VQ-Fusion improved the performance (F1-score) of HAMLET and MuMu models to **87.69%** from **83.89%**. Additionally, VQ-Fusion improved the performance (F1-score) of MuMu models to **87.83%** from **87.50%**. This result is crucial as it underscores the adaptability and effectiveness of the VQ-Fusion model in real-world scenarios, enhancing multimodal representation learning across different domains and datasets.

---

> ### Author Response · Authors · 2023-11-19
> **Clarification of domain gap and impact of EQA-MX dataset (Part 3/4)**
>
> **Reviewer Comment:**
>
> > Synthetic datasets contain a significant domain gap to reality. Real scenes have more noise, clutter, lighting variations, etc and experiments purely in simulations and the conclusions obtained from them cannot be guaranteed to hold in the real world as well.
>
> **Response:** In the previous response, we referred to an additional experimental evaluation of our proposed approach on a real-world dataset. This experimental result suggests the adaptability of our proposed approach to extract salient multimodal representation in real-world settings. However, we acknowledge the concern about the domain gap between our synthetic EQA-MX dataset and real-world scenarios. To mitigate this, in our EQA-MX dataset, we have introduced diverse environmental settings and varied human nonverbal cues such as gaze and gestures, aiming to mimic the complexity and variability found in real-world interactions. This includes introducing new environments and interaction dynamics that were not present in the CAESAR simulator. Still, our synthetic dataset may not capture all the real-world interactions and scene complexities. Along this line, in our ongoing project, we are working to collect a diverse real-world embodied dataset. Till now, the data collection phase involved 392 sessions split between home and lab environments. A total of 13,990 interactions were recorded, with 3,176 occurring at home and 10,814 in the lab. The total video time recorded was 17.62 hours, with 4.14 hours coming from at-home sessions and 13.48 hours from lab sessions. We believe that EQA-MX and our upcoming real-world embodied dataset introduce interesting research directions for transferring models from the synthetic to the real world.
>
>
> Additionally, we would like to clarify the main objective of developing a large-scale synthetic dataset. The primary objective of our dataset is to provide a comprehensive dataset for systematically examining the impact of multimodal interactions—combining nonverbal, verbal, and visual cues—in embodied AI. While the EQA-MX dataset is a synthetic environment, it is designed to encompass a broad spectrum of interactions, replicating the complexity and diversity of real-world scenarios to a significant extent. Controlling different aspects of human interactions is difficult in real-world environments. For example, collecting data from multiple visual and verbal perspectives is costly and time-consuming, whereas we generated a huge amount of training and evaluation data to assess the model's capabilities to comprehend instruction from multiple visual and verbal perspectives.
>
> Finally, the findings from our work present a novel direction in studying and developing models capable of understanding human-embodied interactions in a controlled environment. While we acknowledge the limitations of not including real-world datasets in our current study, the insights gained from our research are valuable in guiding the development of more robust and generalizable models. In future work, we aim to extend our methodology to real-world datasets, further bridging the gap and enhancing the applicability of our findings to practical scenarios. We plan to extend the EQA-MX dataset by collecting real-world human interactions.

---

> ### Author Response · Authors · 2023-11-19
> **Additional ablation study on varying visual views (Part 4/4)**
>
> **Reviewer Comment:**
>
> > While the authors show the value of VQ in Table 3, the value of using multiple views has not been ablated. How much does having visual information from multiple views help in EQA? It seems obvious that it should, especially if some views are likely to have occlusions and ambiguities while other are not, but it would have been nice to see its effect quantitatively.
>
> **Response:** Thank you very much for pointing out the importance of conducting an evaluation by varying the visual views. We conducted the experimentation by ablating the visual views, which are included in the supplementary (Section 3.1, Table 1). We varied the visual perspectives during training and testing by using different camera views (ego, exo, and top) to capture the nonverbal interactions. We used the CLIP model to learn EQA tasks involving verbal utterances and visual views. The results suggest that models trained using multiple visual perspectives perform better than models trained using a single visual perspective (Table 1: Bottom). The reasoning behind this performance improvement is that models using multiple visual views can learn generalized multiview representations, which can improve the performance at inference time when visual views are varied. Additionally, as the reviewer pointed out, multiple visual views provide complementary view representations that aid in extracting comprehensive information to address occlusion and ambiguities in the scene and thus improve the performance across tasks.

---

> > ### Comment · Reviewer_7hsY · 2023-11-22
> > **Thank you, for your response.**
> >
> > I thank the authors of their detailed responses to both my questions. I am satisfied with their responses and have no further pending doubts. I wish the authors good luck!

---

> > > ### Author Response · Authors · 2023-11-23
> > > **Thank you Reviewer 7hsY**
> > >
> > > Thank you for your encouraging words and for taking the time to review our work thoroughly. We're glad to have addressed your questions satisfactorily. Your support and well wishes are greatly appreciated!

---

### Meta-Review · Area_Chair_8o4y · 2023-12-17

**Metareview:**

The authors introduce a new synthetic dataset for embodied question answering that includes additional signals (e.g. gaze and gesture) that are well established signals for human robot interaction in the real world and within the cogsci community for communication and language acquisition.  Reviewers were both happy with the extensiveness of the introduced dataset and the model + ablations included.  This included the discussion of codebooks and multi-view representations.

**Justification For Why Not Higher Score:**

In a word, transfer.  It's unclear right now whether models trained on this dataset will fit to artifacts of the simulation that inhibit transfer to real settings and whether that will limit the impact of the phenomena that appear to be captured here.

**Justification For Why Not Lower Score:**

Reviewers uniformly appreciate the contributions of this work and the vision that it lays out for the field.  The phenomena are important as both the contributions (data and model) significant.  This work will hopefully help motivate the community towards a richer understanding of EQA that is more truthful to HRI and human experience.

---

### Decision · Program_Chairs · 2024-01-16

Accept (spotlight)